# Learning to Boost Resilience of Complex Networks via Neural Edge Rewiring

**Shanchao Yang**  *shanchaoyang@link.cuhk.edu.cn*
*School of Data Science*
*The Chinese University of Hong Kong, Shenzhen*

**Kaili Ma**  *klma@cse.cuhk.edu.hk*
*Department of Computer Science and Engineering*
*The Chinese University of Hong Kong*

**Baoxiang Wang**  *bxiangwang@cuhk.edu.cn*
*School of Data Science*
*The Chinese University of Hong Kong, Shenzhen*

**Tianshu Yu**  *yutianshu@cuhk.edu.cn*
*School of Data Science*
*The Chinese University of Hong Kong, Shenzhen*

**Hongyuan Zha**  *zhahy@cuhk.edu.cn*
*School of Data Science*
*The Chinese University of Hong Kong, Shenzhen*
*Shenzhen Institute of Artificial Intelligence and Robotics for Society*

**Reviewed on OpenReview:** *https://openreview.net/forum?id=moZvOx5cxe*

## Abstract

The resilience of complex networks refers to their ability to maintain functionality in the face of structural attacks. This ability can be improved by performing minimal modifications to the network structure via degree-preserving edge rewiring-based methods. Existing learning-free edge rewiring methods, although effective, are limited in their ability to generalize to different graphs. Such a limitation cannot be trivially addressed by existing graph neural networks (GNNs)-based learning approaches since there is no rich initial node features for GNNs to learn meaningful representations. In this work, inspired by persistent homology, we specifically design a variant of GNN called FireGNN to learn meaningful node representations solely from graph structures. We then develop an end-to-end inductive method called ResiNet, which aims to discover **resi**lient **net**work topologies while balancing network utility. ResiNet reformulates the optimization of network resilience as a Markov decision process equipped with edge rewiring action space. It learns to sequentially select the appropriate edges to rewire for maximizing resilience. Extensive experiments demonstrate that ResiNet outperforms existing approaches and achieves near-optimal resilience gains on various graphs while balancing network utility.

## 1 Introduction

Network systems, such as infrastructure systems and supply chains, are susceptible to malicious attacks, which necessitates addressing their vulnerability through the concept of *network resilience*. Network resilience serves as a metric to assess the ability of a network system to withstand failures and defend itself against attacks (Schneider et al., 2011). Figure 1 visualizes this scenario that the failures of a dozen of nodes could jeopardize the connectivity and utility of the EU power network. Maintaining network resilience is crucial in

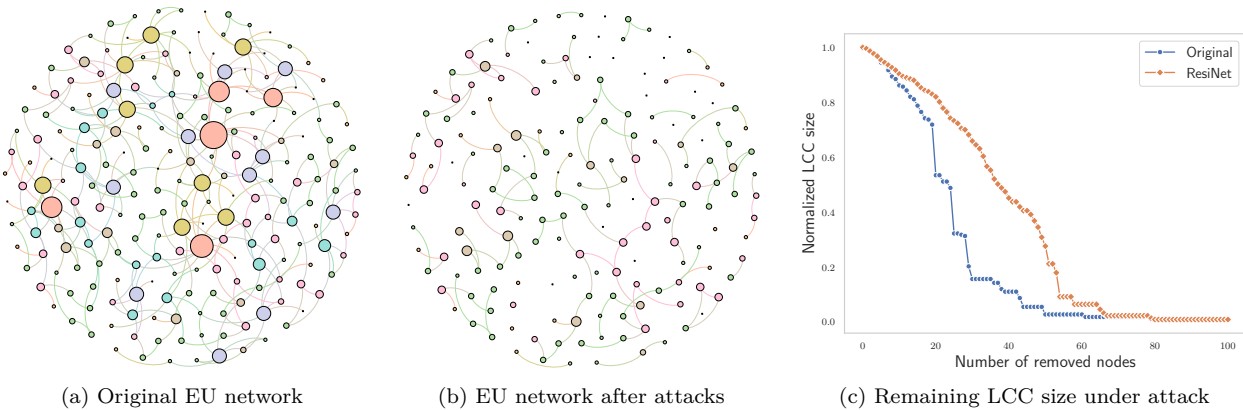

(a) Original EU network           (b) EU network after attacks          (c) Remaining LCC size under attack

Figure 1: The EU power network under the adaptive degree-based attack which removes the most critical node recursively with (a) original EU network with 217 nodes, (b) remaining EU network after a series of attacks on 40 nodes, and (c) the change of the normalized size of the largest connected component (LCC). The node size is proportional to its degree and the node color is given by DBSCAN (Ester et al., 1996).

ensuring that networked systems continue to function and provide an acceptable level of utility, even when confronted with natural disasters or targeted attacks. Consequently, the study of the resilience of complex networks has found widespread applications in various fields, including ecology (Sole & Montoya, 2001), biology (Motter et al., 2008), economics (Haldane & May, 2011), and engineering (Albert et al., 2004).

To enhance network resilience, numerous learning-free optimization methods have been proposed, typically falling into heuristic-based (Schneider et al., 2011; Chan & Akoglu, 2016; Yazıcıoğlu et al., 2015; Rong & Liu, 2018) and evolutionary computation (Zhou & Liu, 2014) categories. These methods aim to improve the resilience of complex networks by making minimal modifications to graph topologies using a degree-preserving atomic operation known as *edge rewiring* (Schneider et al., 2011; Chan & Akoglu, 2016; Rong & Liu, 2018). Specifically, for a given graph $G = (V, E)$ and two existing edges $AC$ and $BD$, an edge rewiring operation alters the graph structure by removing $AC$ and $BD$ and adding $AB$ and $CD$, where $AC, BD \in E$ and $AB, CD, AD, BC \notin E$. Edge rewiring possesses several advantageous properties when compared to simple addition or deletion of edges although edge addition generally outperforms edge rewiring (Beygelzimer et al., 2005). First, edge rewiring preserves node degrees, ensuring capacity constraints are not violated and incurring less network costs (Freitas et al., 2022). Second, edge rewiring minimizes utility degradation in terms of graph Laplacian measurement and thus preserves important graph properties, which may not be the case with edge addition or deletion (Jaume et al., 2020; Ma et al., 2021).

Despite their success, learning-free methods share the following limitations:

- *Transduction.* Existing methods for selecting edges for rewiring are transductive, meaning that they search for robust topologies specific to each individual graph instance. This search procedure is performed independently for each graph and does not generalize across graphs, even if the graphs only differ slightly in structure.

- *Local optimality.* Combinatorially choosing two edges to rewire in order to achieve globally optimal resilience is an NP-hard problem (Mosk-Aoyama, 2008). Previous studies rely primarily on greedy algorithms, resulting in local optimality in practice (Chan & Akoglu, 2016).

- *Utility Loss.* Rewiring operation in network resilience optimization may result in significant degradation of the network utility, potentially compromising the network's overall functionality.

To the best of our knowledge, there is currently no learning-based inductive method for optimizing network resilience via edge rewiring. One of the key challenges lies in the fact that many network science tasks, including resilience optimization, often involve pure network topologies without rich node features. Learning paradigms based on Graph Neural Networks (GNNs) have demonstrated their effectiveness in solving a wide

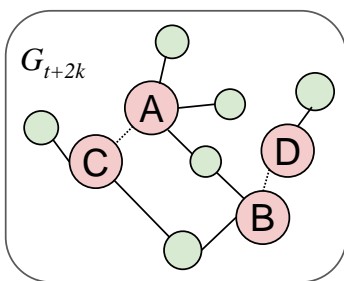 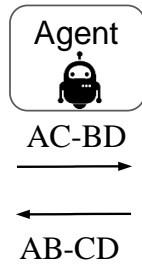 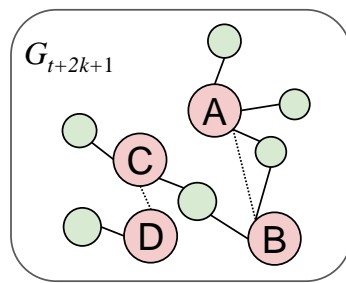

Figure 2: Action backtracking in successive edge rewirings. The graph $G_{t+2k}$ is shown, where $AC$ and $BD$ represent existing edges. To preserve the node degree, at step $t + 2k$, one edge rewiring removes $AC$ and $BD$ from $G_{t+2k}$ and introduces $AB$ and $CD$, resulting in the new graph $G_{t+2k+1}$. However, in cases where GNNs fail to provide distinguishable edge representations on graphs lacking rich features, the agent may select $AB$ and $CD$ for rewiring at step $t + 2k + 1$, leading to a cycle of action backtracking between $G_{t+2k}$ and $G_{t+2k+1}$.

range of graph-related tasks when rich features are available in an inductive manner (Li et al., 2018; Joshi et al., 2019; Khalil et al., 2017; Nazari et al., 2018; Peng et al., 2020). However, it remains challenging to adapt these approaches to tasks that rely solely on topological structures, particularly those that require distinguishable node/edge representations for sequentially constructing a solution. For instance, Boffa et al. (2022) demonstrated a significant performance degradation of GNNs when solving the Traveling Salesman Problem (TSP) without incorporating node coordinate features. Similarly, we have empirically observed that the popular combination of GNNs and reinforcement learning (RL) fails to optimize network resilience via edge rewiring, even though a similar framework has been successfully employed to improve the resilience via edge additions (Darvariu et al., 2021; 2023). Unlike edge addition, which can alter the node degree, edge rewiring preserves the node degree, leading to a graph task with fewer rich features. Therefore, the RL agent gets trapped in an undesired infinite action backtracking loop without meaningful edge representations, as illustrated in Figure 2. A more detailed analysis can be found in Appendix D.

Therefore, devising a novel graph neural network (GNN) that can effectively handle network resilience optimization tasks without relying on rich features is a challenging endeavor. In this study, we address the aforementioned limitation of GNNs in modeling graphs without rich features. We propose the first *inductive* learning-based method for discovering resilient networks through successive edge rewiring operations. To accomplish this, we introduce a specialized variant of GNN called **Fi**lt**r**ation **e**nhanced **GNN** (FireGNN). FireGNN draws inspiration from persistent homology and persistence diagrams (Edelsbrunner & Harer, 2008; Aktas et al., 2019; Hofer et al., 2020; Horn et al., 2022). Persistent homology is a mathematical framework that measures the lifetime of specific topological properties within a simplicial complex as simplices are added or removed. The sequence of subcomplexes constructed during this process, known as filtration, provides valuable information about the network's resilience quality (Horak et al., 2009). Motivated by this, FireGNN generates a filtration by iteratively removing the node with the highest degree from the original graph, resulting in a series of subgraphs. By employing this filtration process, FireGNN learns to aggregate node representations from each subgraph, enabling the acquisition of meaningful representations. This innovative approach enhances the representative power of FireGNN and addresses the challenge of modeling graphs without rich features. Due to the ability of FireGNN to learn these meaningful representations, ResiNet successfully avoids becoming trapped in undesired infinite action backtracking loops, a common failure case observed in other GNNs.

The main contributions of this paper can be summarized as follows:

1) We propose ResiNet, the first learning-based method designed to enhance network resilience via edge rewiring without relying on rich node features. ResiNet employs an inductive approach to preserve degrees while minimizing utility loss during the resilience optimization process. It formulates resilience optimization as a sequential decision-making process for neural edge rewirings. Extensive experiments demonstrate that ResiNet achieves near-optimal resilience while effectively balancing network utilities, outperforming existing approaches by a significant margin.

2) FireGNN, our technical innovation serving as the graph feature extractor, can learn meaningful representations from pure topological structures. FireGNN provides sufficient training signals to train an RL agent to learn successive edge rewiring operations inductively.

## 2 Related Work

**GNNs for graph-related tasks with rich features.** GNNs are powerful tools to learn from relational data with rich features, providing meaningful representations for downstream tasks. Several successful applications using GNNs as backbones include node classification (Kipf & Welling, 2017; Hamilton et al., 2017), link prediction (Li et al., 2020a; Kipf & Welling, 2017), graph property estimation (Xu et al., 2019; Kipf & Welling, 2017; Li et al., 2020a; Bodnar et al., 2021), and combinatorial problems on graphs (e.g., TSP (Li et al., 2018; Joshi et al., 2019; Khalil et al., 2017; Hudson et al., 2022), vehicle routing problem (Nazari et al., 2018; Peng et al., 2020), graph matching (Yu et al., 2021) and adversarial attack on GNNs (Ma et al., 2021; Dai et al., 2018)). However, till now, it remains unclear how to adapt GNNs to graph tasks without rich feature (Zhu et al., 2021) like the resilience optimization task that we focus on. Current topology-based GNNs like TOGL (Horn et al., 2022) rely on distinct node features for calculating the filtration, while our proposed FireGNN addresses this by creating a sequential-related filtration and learning to aggregate them.

**Network resilience.** Modern network systems are threatened by various malicious attacks, such as the destruction of critical nodes, critical connections and critical subset of the network via heuristics/learning-based attack (Fan et al., 2020; Iyer et al., 2013; Grassia et al., 2021; Fan et al., 2020). Network resilience was proposed and proved as a suitable measurement for describing the robustness and stability of a network system under such attacks (Schneider et al., 2011). Around optimizing network resilience, various defense strategies have been proposed to protect the network functionality from crashing and preserve network's topologies to some extent. Commonly used manipulations of defense include adding additional edges (Li et al., 2019; Carchiolo et al., 2019; Darvariu et al., 2021; 2023), protecting vulnerable edges (Wang et al., 2014) and rewiring two edges (Schneider et al., 2011; Chan & Akoglu, 2016; Buesser et al., 2011). Among these manipulations, edge rewiring fits well to real-world applications as it induces fewer functionality changes to the original network and does not impose additional loads to the vertices (degree-preserving) (Schneider et al., 2011; Rong & Liu, 2018; Yazıcıoğlu et al., 2015). By now, there has been no learning-based inductive edge rewiring strategy for the resilience task.

**Extended related work.** The related work on *network utility*, *graph structure learning*, *graph rewiring*, *multi-views graph augmentation for GNNs* and *deep graph generation* is deferred to Appendix A.

## 3 Problem Definition

An undirected graph is defined as $G = (V, E)$, where $V = \{1, 2, \ldots, N\}$ is the set of $N$ nodes, $E$ is the set of $M$ edges, $A \in \{0, 1\}^{N \times N}$ is the adjacency matrix, and $F \in \mathbb{R}^{N \times d}$ is the $d$-dimensional node feature matrix[1]. The degree of a node is defined as $d_i = \sum_{j=1}^{N} A_{ij}$, and a node with degree 0 is called an isolated node. Let $\mathbb{G}_G$ denote the set of graphs with the *same* node degrees as $G$.

Let $\mathbb{G}_G$ denote the set of graphs with the same node degrees as $G$. Given the network resilience metric $\mathcal{R}(G)$ (Schneider et al., 2011) and the utility metric $\mathcal{E}(G)$ (Latora & Marchiori, 2003), the objective of boosting the resilience of $G$ is to find a target graph $G^{\star} \in \mathbb{G}_G$, which maximizes the network resilience while balancing the network utility. Formally, the problem of maximizing the resilience of complex networks is formulated as

$$G^{\star} = \arg\max_{G^{'} \in \mathbb{G}_G} (1 - \alpha) \cdot \mathcal{R}(G^{'}) + \alpha \cdot \mathcal{E}(G^{'}),$$

where $\alpha \in \mathbb{R}$ is the scalar weight that balances the resilience and the utility.

---

[1]For a graph with pure topological structure, node feature matrix is not available.

To satisfy the constraint of preserving degree, edge rewiring is the default atomic operation for obtaining new graphs $G'$ from $G$. Combinatorially, a total of $T$ successive steps of edge rewiring has the complexity of $O(E^{2T})$.

Following the conventional setting in network science, resilience metrics used in our experiments include graph connectivity-based (Schneider et al., 2011) and spectrum-based measurements (adjacency matrix spectrum and Laplacian matrix spectrum). Utility metrics consist of global efficiency and local efficiency (Latora & Marchiori, 2001). The details of metrics are presented as follows.

**Resilience metrics**  Two kinds of resilience metrics are considered:

- The graph connectivity-based resilience measurement is defined as (Schneider et al., 2011)

$$\mathcal{R}(G) = \frac{1}{N} \sum_{q=1}^{N} s(q) \,,$$

  where $s(q)$ is the fraction of nodes in the largest connected remaining graph after removing $q$ nodes from $G$ according to certain attack strategy. The range of possible values of $\mathcal{R}$ is $[1/N, 1/2]$, where these two extreme values correspond to a star network and a fully connected network, respectively.

- The algebraic connectivity ($\mathcal{AC}$) is the second smallest eigenvalue of the Laplacian matrix of $G$.

**Utility metrics**  The global and local communication efficiency are used as two representative measurements of the network utility, which are widely applied across diverse applications of network science, such as transportation and communication networks (Latora & Marchiori, 2003).

The average efficiency of a network $G$ is defined inversely proportional to the average over pairwise distances (Latora & Marchiori, 2001) as

$$E(G) = \frac{1}{N(N-1)} \sum_{i \neq j \in V} \frac{1}{d(i,j)} \,,$$

where $d(i,j)$ is the length of the shortest path between a node $i$ and another node $j$.

Based on the average efficiency, the global efficiency and local efficiency are defined as

- The global efficiency of a network $G$ is defined as (Latora & Marchiori, 2001)

$$E_{global}(G) = \frac{E(G)}{E(G^{ideal})} \,,$$

  where $G^{ideal}$ is the "ideal" fully-connected graph on $N$ nodes and the range of $E_{global}(G)$ is $[0, 1]$.

- The local efficiency of a network $G$ measures a local average of pairwise communication efficiencies and is defined as (Latora & Marchiori, 2001)

$$E_{local}(G) = \frac{1}{N} \sum_{i \in V} E(G_i) \,,$$

  where $G_i$ is the local subgraph including only of a node $i$'s one-hop neighbors, but not the node $i$ itself. The range of $E_{local}(G)$ is $[0, 1]$.

## 4  Proposed Approach: ResiNet

In this section, we formulate the task of boosting network resilience via edge rewiring as a reinforcement learning task by learning to select two edges and rewire them successively. We first present the graph resilience-aware environment design and describe our innovation FireGNN in detail. Finally, we present the graph policy network that guides the edge selection and rewiring process.

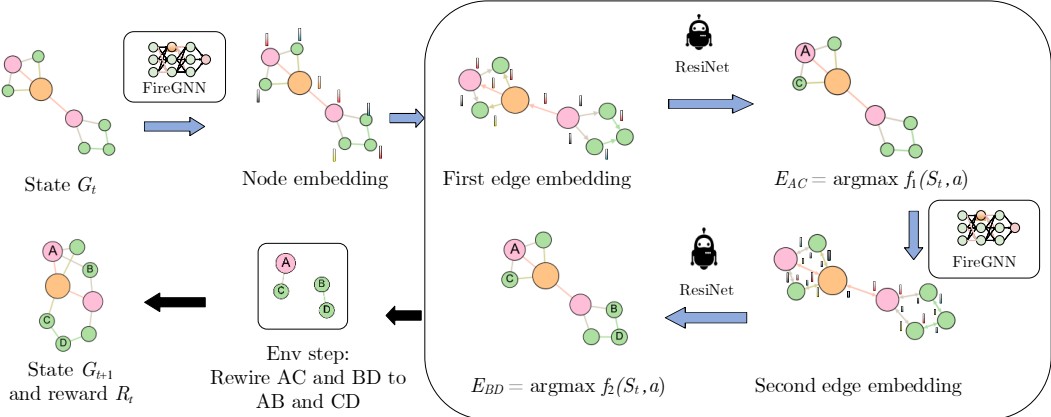

Figure 3: Overview of the architecture of ResiNet to select two edges for edge rewiring.

## 4.1 Boosting Network Resilience via Edge Rewiring as Markov Decision Process

To satisfy the constraint of preserving the node degree, the resilience optimization of a given graph is based on edge rewiring. We formulate the network resilience optimization problem via successive edge rewiring operations into the Markov decision process (MDP) framework. The Markov property denotes that the graph obtained at time step $t+1$ relies only on the graph at time step $t$ and the rewiring operation, reducing the complexity from original $O(E^{2T})$ to $O(TE^2)$. Then we further reduce the complexity to $O(TE)$ by designing an autoregressive edge selection module shown as follows.

As shown in Figure 3, ResiNet performs the resilience optimization in an auto-regressive step-wise way through a sequence of edge rewiring actions. Given an input graph, the agent first decides whether to terminate or not. If not, it selects one edge from the graph to remove, receives the very edge it just selected as the auto-regression signal, and then selects another edge to remove. Four nodes of these two removed edges are re-combined, forming two new edges to be added to the graph. The optimization process repeats until the agent decides to terminate.

After formulating the resilience optimization task as a MDP, reinforcement learning (RL) can be employed as a viable solution approach. RL enables an agent to learn optimal decision-making by interacting with an environment. The environment can be modeled as an MDP, comprising a set of states, actions, transition probabilities, and rewards. At each time step, the agent observes the current state, selects an action, and receives feedback in the form of a reward signal and the subsequent state. The ultimate objective of the agent is to learn a policy, which is a mapping from states to actions, that maximizes the cumulative reward over time. Herein, we provide a detailed description of the design for the state representation, action selection, transition dynamics, and reward structure.

**State.** The fully observable state is formulated as $S_t = G_t$, where $G_t$ is the current input graph at step $t$. The widely-used node degree feature cannot significantly benefit the network resilience optimization of a single graph due to the degree-preserving rewiring. Therefore, we construct node features for each input graph to aid the transductive learning and inductive learning, including

- The distance encoding strategy (Li et al., 2020b). Node degree feature is a part of it.

- The 8-dimensional position embedding originating from the Transformer (Vaswani et al., 2017) as the measurement of the vulnerability of each node under attack. If the attack order is available, we can directly encode it into the position embedding. If the attack order is unknown, node degree, node betweenness, and other node priority metrics can be used for approximating the node importance in practice. In our experiments, we used the adaptive node degree for the position embedding.

**Action.** ResiNet is equipped with a node permutation-invariant, variable-dimensional action space. Given a graph $G_t$, the action $a_t$ is to select two edges and the rewiring order. The agent first chooses an edge $e_1 = AC$ and a direction $A \rightarrow C$. Then conditioning on the state, $e_1$, and the direction the agents chooses an edge $e_2 = BD$ such that $AB, CD, AD, BC \notin E$ and a direction $B \rightarrow D$. The heads of the two edges reconnect as a new edge $AB$, and so does the tail $CD$. Although $G_t$ is undirected, we propose to consider the artificial edge directions, which effectively avoids the redundancy in representing action space since $A \rightarrow C$, $B \rightarrow D$ and $C \rightarrow A$, $D \rightarrow B$ refer to the same rewiring operation. The choice of the direction of $e_1$ is randomized (this randomized bit is still an input of choosing $e_2$). Therefore, our proposed action space effectively reduces the size of the original action space by half and still leads to a complete action space. In this way, the action space is the set of all feasible pairs of $(e_1, e_2) \in E^2$

**Transition dynamics.** The formulation of the action space implies that if the agent does not terminate at step $t$, the selected action must form an edge rewiring. This edge rewiring is executed by the environment, and the graph transits to the new graph.

Note that in some other work, infeasible operations are also included in the action space (to make the action space constant through the process) (You et al., 2018; Trivedi et al., 2020). This reduces the sample efficiency and causes biased gradient estimations (Huang & Ontañón, 2020). ResiNet takes advantage of the state-dependent variable action space composed of only feasible operations.

**Reward.** ResiNet aims to optimize resilience while balancing the utility, forming a complicated and possibly unknown objective function. Despite this, by Wakuta (1995), an MDP that maximizes a complicated objective is up to an MDP that maximizes the linear combination of resilience and utility for some coefficient factor. This fact motivates us to design the reward as the step-wise gain of such a linear combination as

$$R_t = (1 - \alpha) \cdot (\mathcal{R}(G_{t+1}) - \mathcal{R}(G_t)) + \alpha \cdot (\mathcal{E}(G_{t+1}) - \mathcal{E}(G_t)) ,$$

where $\mathcal{R}(G)$ and $\mathcal{E}(G)$ are the resilience function and the utility function, respectively. The cumulative reward $\sum_{t=0}^{T-1} R_t$ up to time $T$ is then the total gain of such a linear combination.

### 4.2 FireGNN

Motivated by graph filtration in persistent homology (Edelsbrunner & Harer, 2008), we design the **fil**t**r**ated graph **e**nhanced **GNN** termed FireGNN to model graphs without rich features, or even with only topology. For a given input graph $G$, FireGNN transforms $G$ from the static version to a temporal version consisting of a sequence of subgraphs, by repeatedly removing the node with the highest degree. Observing a sequence of nested subgraphs of $G$ grants FirGNN the capability to observe how $G$ evolves towards being empty. Then FireGNN aligns and aggregates the node, edge, and graph embedding from each subgraph, leading to meaningful representations in node, edge, and graph levels. Formally, the filtration in FireGNN is constructed as

$$G^{(k-1)} = G^{(k)} - v_k, \quad v_k = \underset{v_i \in G^{(k)}}{\operatorname{argmax}} \operatorname{DEGREE}(v_i)$$

$$(V, \emptyset) = G^{(0)} \subset G^{(1)} \subset \cdots \subset G^{(N)} = G$$

$$\tilde{G} = [G^{(0)}, G^{(1)}, \ldots, G^{(N)}] ,$$

where $G^{(k)}$ denotes the remaining graph after removing $N - k$ nodes with highest node degrees, $v_k$ denotes the node with highest degree in current subgraph $G^{(k)}$, $\operatorname{DEGREE}(\cdot)$ measures the node degree, $G^{(N)}$ is the original graph, and $G^{(0)}$ contains no edge. The sequence of the nested subgraphs of $G$ is termed the filtrated graph $\tilde{G}$. We illustrate the filtration process on a toy dataset in Figure 4.

**Node embedding.** Regular GNN only operates on the original graph $G$ to obtain the node embedding for each node $v_i$ as $h(v_i) = \phi(G^{(N)} = G)_i$, where $\phi(\cdot)$ denotes a standard GNN model. In FireGNN, by using the top $K + 1$ subgraphs in a graph filtration, the final node embedding $h(v_i)$ of $v_i$ is obtained by

$$h(v_i) = \operatorname{AGG}_N \left( h^{(N-K)}(v_i), \ldots, h^{(N-1)}(v_i), h^{(N)}(v_i) \right) ,$$

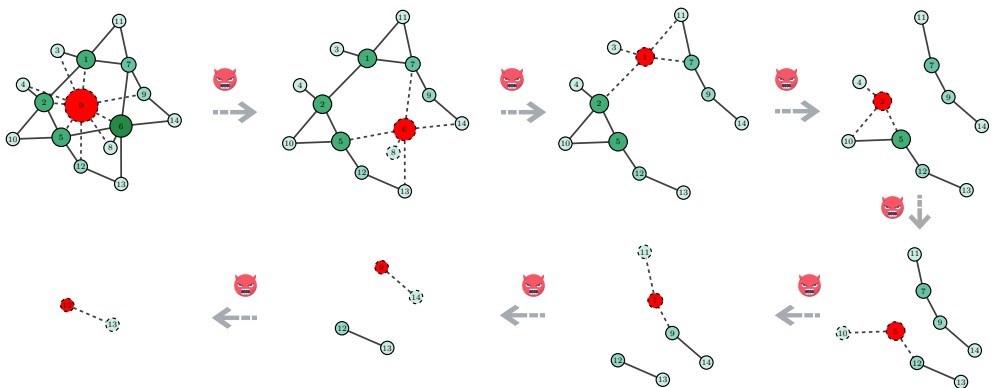

Figure 4: Filtration Process in FireGNN on BA-15. The original graph is decomposed into sequential-related sub-graphs.

where $\text{AGG}_N(\cdot)$ denotes a node-level aggregation function, $h^{(k)}(v_i)$ is the node embedding of $i$ in the $k$-th subgraph $G^{(k)}$ obtained by passing $G$ to a backbone GNN, and $K \in [N]$. In practice, $h^{(k)}(v_i)$ is discarded when calculating $h(v_i)$ if $v_i$ is isolated or not included in $G^{(k)}$.

**Edge embedding.** The directed edge embedding $h^{(k)}(e_{ij})$ of the edge from node $i$ to node $j$ in each subgraph is obtained by combining the embeddings of the two end vertices in $G^{(k)}$ as

$$h^{(k)}(e_{ij}) = m_f \left( \text{AGG}_{N \rightarrow E} \left( h^{(k)}(v_i), h^{(k)}(v_j) \right) \right) ,$$

where $\text{AGG}_{N \rightarrow E}(\cdot)$ denotes an aggregation function for obtaining edge embedding from two end vertices (typically chosen from `min`, `max`, `sum`, `difference`, and `multiplication`). $m_f(\cdot)$ is a multilayer perceptron (MLP) model that ensures the consistence between the dimensions of edge embedding and graph embedding.

The final embedding of the directed edge $e_{ij}$ of the filtrated graph $\tilde{G}$ is given by

$$h(e_{ij}) = \text{AGG}_E \left( h^{(N-K)}(e_{ij}), \ldots, h^{(N-1)}(e_{ij}), h^{(N)}(e_{ij}) \right) ,$$

where $\text{AGG}_E(\cdot)$ denotes an edge-level aggregation function.

**Graph embedding.** With the node embedding $h^{(k)}(v_i)$ of each subgraph $G^{(k)}$ available, the graph embedding $h^{(k)}(G)$ of each subgraph $G^{(k)}$ is calculated by a readout functions (e.g., `mean`, `sum`) on all non-isolated nodes in $G^{(k)}$ as

$$h^{(k)}(G) = \text{READOUT} \left( h^{(k)}(v_i) \right) \forall v_i \in G^{(k)} \text{ and } d_i^{(k)} \geq 0 .$$

The final graph embedding of the filtrated graph $\tilde{G}$ is given by

$$h(G) = \text{AGG}_G \left( h^{(N-K)}(G), \ldots, h^{(N-1)}(G), h^{(N)}(G) \right) ,$$

where $\text{AGG}_G(\cdot)$ denotes a graph-level aggregation function.

### 4.3 Edge Rewiring Policy Network

Having presented the details of the graph resilience environment and FireGNN, in this section, we describe the policy network architecture of ResiNet in detail, which learns to select two existing edges for rewiring at each step. At time step $t$, the policy network uses FireGNN as the graph extractor to obtain the directed edge embedding $h(e_{ij}) \in \mathbb{R}^{2|E| \times d}$ and the graph embedding $h(G) \in \mathbb{R}^d$ from the filtrated graph $\tilde{G}_t$, and outputs an action $a_t$ representing two selected rewired edges, leading to the new state $G_{t+1}$ with reward $R_t$.

To be inductive, we adapt a special autoregressive node permutation-invariant dimension-variable action space to model the selection of two edges from graphs with arbitrary sizes and permutations. The detailed

mechanism of obtaining the action $a_t$ based on edge embedding and graph embedding is presented as follows, further reducing the complexity from $O(TE^2)$ to $O(TE)$.

**Auto-regressive latent edge selection.** An edge rewiring action $a_t$ at time step $t$ involves the prediction of the termination probability $a_t^{(0)}$ and the selection of two edges ($a_t^{(1)}$ and $a_t^{(2)}$) and the rewiring order. The action space of $a_t^{(0)}$ is binary, however, the selection of two edges imposes a huge action space in $O(|E|^2)$, which is too expensive to sample from even for a small graph. Instead of selecting two edges simultaneously, we decompose the joint action $a_t$ into $a_t = (a_t^{(0)}, a_t^{(1)}, a_t^{(2)})$, where $a_t^{(1)}$ and $a_t^{(2)}$ are two existing edges which do not share any common node (recall that $a_t^{(1)}$ and $a_t^{(2)}$ are directed edges for an undirected graph). Thus the probability of $a_t$ is

$$\mathbb{P}(a_t|s_t) = \mathbb{P}(a_t^{(0)}|s_t)\mathbb{P}(a_t^{(1)}|s_t, a_t^{(0)})\mathbb{P}(a_t^{(2)}|s_t, a_t^{(0)}, a_t^{(1)}).$$

**Predicting the termination probability.** The first policy network $\pi_0(\cdot)$ takes the graph embedding as input and outputs the probability distribution of the first action that decides to terminate or not as $\mathbb{P}(a_t^{(0)}|s_t) = \pi_0(h(G))$, where $\pi_0(\cdot)$ is implemented by a two-layer MLP. Then the probability of the first subaction is given as

$$a_t^{(0)} \sim \text{Bernoulli}(\mathbb{P}(a_t^{(0)}|s_t)) \in \{0, 1\}.$$

**Selecting edges.** If the signal $a_t^{(0)}$ given by the agent decides to continue to rewire, two edges are then selected in an auto-regressive way. The signal of continuing to rewire $a_t^{(0)}$ is input to the selection of two edges as a one-hot encoding vector $l_c$. The second policy network $\pi_1(\cdot)$ takes the graph embedding and $l_c$ as input and outputs a latent vector $l_1 \in \mathbb{R}^d$ to determine the selection of edges. To account for the variability in the action space across different graphs, we employ the pointer network (Vinyals et al., 2015) that offers the advantages of handling variable-sized outputs, flexible action selection, and end-to-end training. The pointer network measures the proximity between $l_1$ and each edge embedding $h(e_{ij})$ in $G$ to obtain the first edge selection probability distribution. Then, to select the second edge, the graph embedding $h(G)$ and the first selected edge embedding $h(e_t^{(1)})$ and $l_c$ are concatenated and fed into the third policy network $\pi_2(\cdot)$. $\pi_2(\cdot)$ obtains the latent vector $l_2$ for selecting the second edge using a respective pointer network. The overall process can be formulated as

$$l_1 = \pi_1([h(G), l_c]), \quad \mathbb{P}(a_t^{(1)}|s_t, a_t^{(0)}) = f_1(l_1, h(e_{ij}))$$
$$l_2 = \pi_2([h(G), h(e_t^{(1)}), l_c]), \quad \mathbb{P}(a_t^{(2)}|s_t, a_t^{(1)}, a_t^{(0)}) = f_2(l_2, h(e_{ij})),$$

where $e_{ij} \in E$ and $\pi_i(\cdot)$ is a two-layer MLP model, $[\cdot, \cdot]$ denotes the concatenation operator, $h(e_t^{(1)})$ is the embedding of the first selected edge at step $t$, and $f_i(\cdot)$ is a pointer network.

## 5 Experiments

In this section, we demonstrate the advantages of ResiNet over existing non-learning-based and learning-based methods in achieving superior network resilience, inductively generalizing to unseen graphs, and accommodating multiple resilience and utility metrics. Moreover, we show that FireGNN can learn meaningful representations from graph data without rich features, while current GNNs fail. *Our implementation is available at `https://github.com/yangysc/ResiNet`.*

### 5.1 Experimental Settings

**Datasets.** Synthetic and real datasets including EU power network (Zhou & Bialek, 2005) and Internet peer-to-peer networks (Leskovec et al., 2007; Ripeanu et al., 2002) are used to demonstrate the performance of ResiNet in transductive and inductive settings. The details of data generation and statistics of the datasets are presented in Appendix B.1. Following the conventional experimental settings, the maximal node size is set to be around 1000 (Schneider et al., 2011), taking into account: 1) the high complexity of selecting two edges at each step is $O(E^2)$; 2) evaluating the resilience metric is time-consuming for large graphs.

Table 1: Resilience optimization algorithm under the fixed maximal rewiring number budget of 20. Entries are in the format of $X(Y)$, where 1) $X$: weighted sum of the graph connectivity-based resilience and the network efficiency improvement (in percentage); 2) $Y$: required rewiring number. ✗ means that the algorithm cannot find a solution in a reasonable time.

| Method | $\alpha$ | BA-15 | BA-50 | BA-100 | BA-500 | BA-1000 | EU | P2P-Gnutella05 | P2P-Gnutella09 |
|---|---|---|---|---|---|---|---|---|---|
| HC | 0 | 26.8 (10) | 30.0 (20) | 24.1 (20) | 6.4 (20) | 66.6 (20) | 19.8 (20) | 6.2 (20) | 8.4 (20) |
|  | 0.5 | 18.6 (11.3) | 22.1 (20) | 14.9 (20) | 5.9 (20) | 16.4 (20) | 16.3 (20) | 5.2 (20) | 7.0 (20) |
| SA | 0 | 21.6 (17.3) | 11.9 (20) | 12.5 (20) | 3.8 (20) | 42.9 (20) | 14.9 (20) | 3.9 (20) | 3.7 (20) |
|  | 0.5 | 16.8 (19.0) | 11.4 (20) | 13.4 (20) | 4.0 (20) | 15.4 (20) | 14.0 (20) | 6.3 (20) | 4.8 (20) |
| Greedy | 0 | 23.5 (6) | 48.6 (13) | 64.3 (20) | ✗ | ✗ | 0.5 (3) | ✗ | ✗ |
|  | 0.5 | 5.3 (15) | 34.7 (13) | 42.7 (20) | ✗ | ✗ | 0.3 (3) | ✗ | ✗ |
| EA | 0 | 8.5 (20) | 6.4 (20) | 4.0 (20) | 8.5 (20) | **174.1** (20) | 8.2 (20) | 2.7 (20) | 0 (20) |
|  | 0.5 | 6.4 (20) | 4.7 (20) | 2.8 (20) | 5.6 (20) | 18.7 (20) | 9.3 (20) | 3.7 (20) | 0.1 (20) |
| DE- | 0 | 13.7 (2) | 0 (1) | 0 (1) | 1.6 (20) | 41.7 (20) | 9.0 (20) | 2.2 (20) | 0 (1) |
| GNN-RL | 0.5 | 10.9 (2) | 0 (1) | 0 (1) | 2.7 (20) | 20.1 (14) | 2.1 (20) | 0 (1) | 1.0 (20) |
| $k$-GNN- | 0 | 13.7 (2) | 0 (1) | 0 (1) | 0 (1) | 8.8 (20) | 4.5 (20) | -0.2 (20) | 0 (1) |
| RL | 0.5 | 6.3 (2) | 0 (1) | 0 (1) | 0 (20) | -24.9 (20) | 4.8 (20) | -0.1 (20) | 0 (1) |
| DIGL- | 0 | 9.8 (2) | 0 (1) | 0 (1) | ✗ | ✗ | 5.9 (20) | ✗ | ✗ |
| RL | 0.5 | 6.3 (2) | 0 (1) | 0 (1) | ✗ | ✗ | 7.2 (20) | ✗ | ✗ |
| SDRF- | 0 | 9.8 (2) | 0 (1) | 0 (1) | ✗ | ✗ | 4.7 (20) | ✗ | ✗ |
| RL | 0.5 | 8.0 (2) | 0 (1) | -4.7 (20) | ✗ | ✗ | 5.3 (20) | ✗ | 55 |
| ResiNet | 0 | **35.3** (6) | **61.5** (20) | **70.0** (20) | **10.2** (20) | 172.8 (20) | **54.2** (20) | **14.0** (20) | **18.6** (20) |
| (ours) | 0.5 | **26.9** (20) | **53.9** (20) | **53.1** (20) | **15.7** (20) | **43.7** (20) | **51.8** (20) | **12.4** (20) | **15.1** (20) |

**Baselines.** We compare ResiNet with existing edge rewiring-based graph resilience optimization algorithms, including learning-free and learning-based algorithms. Learning-free methods (upper half of Table 1) include the hill climbing (HC) (Schneider et al., 2011), the greedy algorithm (Chan & Akoglu, 2016), the simulated annealing (SA) (Buesser et al., 2011), and the evolutionary algorithm (EA) (Zhou & Liu, 2014). Since to our knowledge there is no previous learning-based baseline, we specifically devise five counterparts based on our method by replacing FireGNN with existing well-known powerful GNNs (DE-GNN (Li et al., 2020b), $k$-GNN (Morris et al., 2019), DIGL (Klicpera et al., 2019) and SDRF (Topping et al., 2022)) (lower half of Table 1). The classical GIN model is used as the backbone (Xu et al., 2019). All baselines are trained using the same reward as ResiNet.

The ResiNet's training setup is detailed as follows. Our proposed FireGNN is used as the graph encoder in ResiNet, including a 5-layer defined GIN (Xu et al., 2019) as the backbone. The hidden dimensions for node embedding and graph embedding in each hidden layer are set to 64 and the SeLU activation function is used after each message passing propagate. Graph normalization strategy is adopted to stabilize the training of GNN (Cai et al., 2021). The jumping knowledge network (Xu et al., 2018) is used to aggregate node features from different layers of the GNN. The overall policy is trained by using the highly tuned implementation of proximal policy optimization (PPO) algorithm (Schulman et al., 2017). Several critical strategies for stabilizing and accelerating the training of ResiNet are used, including advantage normalization (Andrychowicz et al., 2021), the dual-clip PPO (the dual clip parameter is set to 10) (Ye et al., 2020), and the usage of different optimizers for policy network and value network. Additionally, since the step-wise reward range is small (around 0.01), we scale the reward by a factor of 10 to facilitate the training of ResiNet. The policy head model and value function model use two separated FireGNN encoder networks with the same architecture. We run all experiments for ResiNet on the platform with two GEFORCE RTX 3090 GPU and one AMD 3990X CPU.

## 5.2 Comparisons to the Baselines

In this section, we compare ResiNet to baselines in optimizing the combination of resilience and utility with weight coefficient $\alpha \in \{0, 0.5\}$. Following conventional setting, the graph connectivity-based metric is used as resilience metric (Schneider et al., 2011) and the global efficiency is used as utility metric (Latora & Marchiori, 2003).

Table 1 records the metric gain and the required number of rewiring operations of different methods under the same rewiring budget. ResiNet outperforms all baselines consistently on all datasets. Note that this

performance may be achieved by ResiNet under a much fewer number of rewiring operations, such as on BA-15 with $\alpha = 0$. In contrast, despite approximately searching for all possible new edges, the greedy algorithm is trapped in a local optimum (as it maximizes the one-step resilience gain) and is too expensive to optimize the resilience of a network with more than 300 nodes. For SA, the initial temperature and the temperature decay rate need to be carefully tuned for each network. EA performs suboptimally with a limited rewiring budget due to the numerous rewiring operations required in the internal process (e.g., the crossover operator). Learning-based methods using existing GNNs coupled with distance encoding cannot learn effectively compared to our proposed ResiNet, supporting our claim about the effectiveness of FireGNN on graphs without rich features.

### 5.3   Ablation Study of ResiNet

In this section, we investigate the impact of coefficient $\alpha$ of the objective on ResiNet and the effect of the filtration order $K$ on FireGNN.

To investigate the impact of the $\alpha$ in the reward function on ResiNet, we run a grid search by varying $\alpha$ from 0 to 1 and summarize the resilience gain, utility gain, and the sum of them in Table 2. Table 2 shows that when we only optimize the resilience with $\alpha = 0$, the utility will degrade. Similarly, the resilience would also decrease if we only optimize the utility with $\alpha = 1$. This suggests a general tradeoff between resilience and utility and is consistent with their definitions. However, despite this tradeoff, we can achieve resilience gain and utility gain simultaneously on BA-15 and BA-50 since the original graph usually does not have the maximum resilience or utility. This incentivizes almost every network conducts such optimization to some extent when feasible.

In FireGNN, the filtration order $K$ of FireGNN determines the total number of subgraphs involved in calculating the final node embedding, edge embedding, and graph embedding. FireGNN degenerates to existing GNNs when the filtration order $K$ is 0. Table 1 validates the effectiveness and necessity of FireGNN. Without FireGNN (other GNNs as the backbone), it is generally challenging for ResiNet to find a positive gain on graphs without rich features since ResiNet cannot learn to select the correct edges with the incorrect edge embeddings. The maximum $K$ of each dataset is recorded in Appendix Table 6, which shows that the maximum $K$ equals the around half size of the graph since we gradually remove the node with the largest degree, leading to a fast graph filtration process. For our experiments, we use the maximum of $K$ for graphs of sizes less than 50 and set $K = 3$ (1) for graphs of sizes larger than 50 (200). To validate that ResiNet is not sensitive to $K$, we run a grid search on several datasets to optimize the resilience by setting $K = 0, 1, 2, 3$. As shown in Appendix Table 4, the resilience is improved significantly with $K > 0$ and ResiNet performs well with $K = 1$ or $K = 2$. In practice, the choice of the filtration order, denoted as $K$, depends on the size of the graph. For small and moderate graphs, it is reasonable to set $K$ to the maximum size of the nodes. This allows for a more comprehensive exploration of the graph's substructures during the filtration process. However, when dealing with larger graphs, computational limitations may arise. In such cases, it is advisable to set $K$ to a smaller value, such as 1 or 2, to mitigate the computational burden while still capturing important subgraph information. This approach strikes a balance between computational efficiency and retaining key insights from the filtration process.

### 5.4   Stability and Generalization

ResiNet leverages our proposed FireGNN to acquire meaningful representations that enhance the resilience of graphs with purely topological structures. By design, ResiNet inherits the stability properties of the backbone GNN employed in FireGNN. As GNNs have been demonstrated to possess permutation equivalence and stability against relative perturbations in the underlying graph structure (Gama et al., 2019), it follows that ResiNet exhibits stability in the face of such perturbations.

To demonstrate the induction of ResiNet, we first train ResiNet on two different datasets (BA-10-30 and BA-20-200), and then evaluate its performance on an individual test dataset. We report the averaged resilience gain for the graphs of the same size for each dataset.

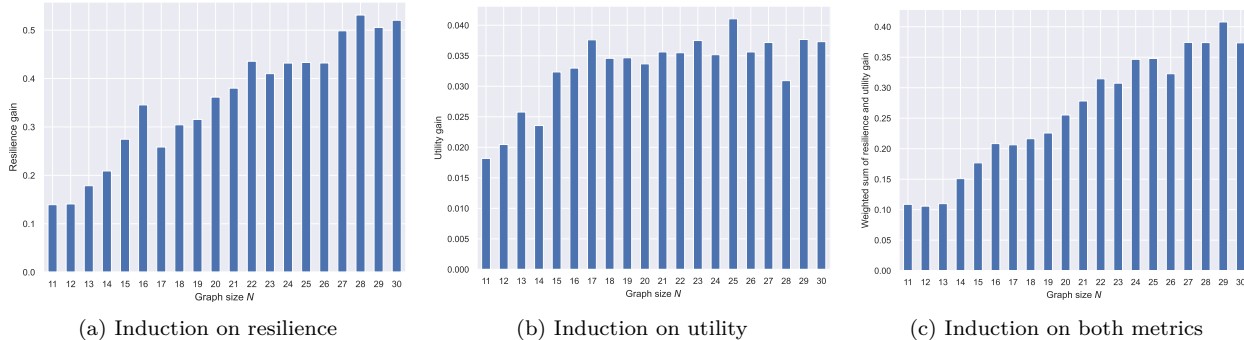

(a) Induction on resilience  (b) Induction on utility  (c) Induction on both metrics

Figure 5: The inductive ability of ResiNet on the test dataset (BA-10-30) when optimizing (a) network resilience, (b) network utility, and (c) their combination.

Table 2: The effect of the coefficient $\alpha$ on ResiNet. The result is shown as percentages.

| Dataset | Gain | 0 | 0.1 | 0.2 | 0.3 | 0.4 | 0.5 | 0.6 | 0.7 | 0.8 | 0.9 | 1 |
|---------|------|-----|-----|-----|-----|-----|-----|-----|-----|-----|-----|-----|
| BA-15 | Resilience | 35.3 | 35.3 | 35.3 | 33.3 | 17.6 | 17.6 | 27.5 | 17.6 | 17.6 | 17.6 | -2.0 |
| | Utility | -5.9 | -3.9 | -3.8 | -2.7 | 1.1 | 1.1 | 0 | 1.1 | 1.1 | 1.1 | 5.4 |
| | Reward | 35.3 | 34.2 | 32.9 | 29.7 | 15.2 | 14.2 | 19.7 | 11.4 | 9.2 | 6.0 | 5.4 |
| BA-50 | Resilience | 56.7 | 51.1 | 42.3 | 48.6 | 53.9 | 59.2 | 51.4 | 50.6 | 48.1 | 39.3 | -19.1 |
| | Utility | -3.6 | 3.4 | -2.1 | -4.0 | -4.2 | -4.2 | -2.6 | -2.2 | -2.1 | 0.5 | 5.5 |
| | Reward | 56.7 | 49.5 | 39.9 | 43.1 | 44.9 | 45.6 | 35.7 | 30.1 | 22.0 | 11.8 | 5.5 |
| BA-100 | Resilience | 75.4 | 74.6 | 74.8 | 76.1 | 72.8 | 72.8 | 75.1 | 75.4 | 74.9 | 71.6 | -11.8 |
| | Utility | -4.0 | -4.6 | -3.9 | -5.1 | -4.2 | -4.2 | -3.8 | -3.7 | -3.5 | -2.5 | 4.8 |
| | Reward | 75.4 | 71.9 | 69.0 | 66.4 | 59.4 | 54.3 | 49.7 | 41.8 | 31.1 | 16.7 | 4.8 |

The performance of ResiNet on BA-10-30 is shown in Figure 5 and the results of other datasets are deferred to Figure 8 in Appendix C. Figure 5 shows a nearly linear improvement of resilience with the increase of graph size, which is consistent with the results in the transductive setting that larger graphs usually have a larger room to improve their resilience. Moreover, we conduct experiments to demonstrate the generalization capabilities of ResiNet in optimizing different utility and resilience metrics, and the details are deferred to Appendix C.

We also use BA-15 as an example to visualize how the values of resilience and utility change as $\alpha$ increases from 0 to 1, as shown in Figure 6. The Pareto frontier line in Figure 6 reveals that our algorithm results in two tradeoff-friendly regions. Specifically, when $\alpha$ falls within the range of 0.6 to 1.0, enhancing network resilience has minimal impact on network utility.

## 5.5 Limitations and Future Work

Inspired by persistent homology, our proposed FireGNN utilizes a filtration process to generate a sequence of subgraphs to improve the representative power of GNNs. Firstly, FireGNN employs a backbone GNN to obtain the embeddings of each subgraph. Subsequently, it learns to aggregate these embeddings from various subgraphs to derive the final node and graph embeddings. This approach demonstrates excellent performance on

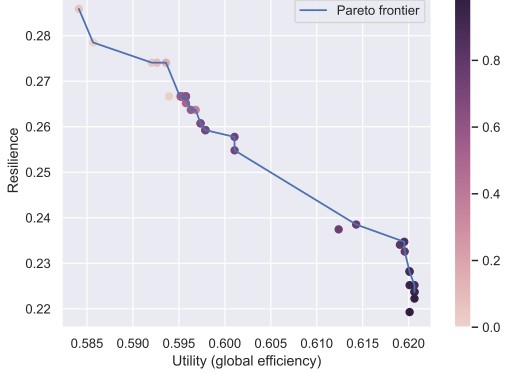

Figure 6: The value changes of graph connectivity-based resilience and global efficiency-based utility as the $\alpha$ increases from 0 to 1 on the dataset BA-15.

graphs of moderate sizes. However, for larger graphs, the computational cost may become prohibitive due to the requirement of calculating and storing all intermediate subgraph embeddings. To address this lim-

itation, it would be valuable to develop an efficient version of the filtration process that avoids redundant GNN computations on subgraphs. A promising avenue is to investigate techniques such as precomputing embeddings (Yan et al., 2023), which can enhance the scalability of FireGNN.

## 6   Conclusion

We have proposed ResiNet, a learning-based inductive method for the discovery of resilient network topologies via edge rewiring with minimal changes to the graph structure. ResiNet is the first inductive edge rewiring-based method that formulates the task of boosting network resilience as an MDP of successive rewiring operations. Our technical innovation, FireGNN, is motivated by persistent homology as the graph feature extractor for handling graphs with only topologies available. FireGNN alleviates the insufficiency of current GNNs (including GNNs more powerful than 1-WL test) on modeling graphs lacking rich features. By decomposing graphs into a sequence of subgraphs and learning to combine the individual representations from each subgraph, FireGNN can learn meaningful representations on the resilience task to provide sufficient gradients for training an RL agent to select correct edges while current GNNs fail due to the infinite action backtracking. Our method is practical as it effectively balances network utility when boosting resilience. FireGNN is potentially general enough to be applied to solve various graph problems without rich features.

## Acknowledgements

We want to thank the reviewers and the editors for their constructive comments during the review process. Baoxiang Wang is partially supported by National Natural Science Foundation of China (62106213, 72150002) and Shenzhen Science and Technology Program (RCBS20210609104356063, JCYJ20210324120011032). Hongyuan Zha is partially supported by Shenzhen Science and Technology Program (JCYJ20210324120011032, ZDSYS20220606100601002) and a grant from Shenzhen Institute of Artificial Intelligence and Robotics for Society.

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

**Appendix**

# A    Extended Related Work

**Network utility.**    Network utility refers to the system's quality to provide a specific service, for example, transmitting electricity in power networks and transmitting packages in routing networks. A popular metric for network utility is the network efficiency Latora & Marchiori (2003). In many previous work, despite that network resilience could be improved, the utility may dramatically drop at the same time Carchiolo et al. (2019); Schneider et al. (2011); Chan & Akoglu (2016); Buesser et al. (2011). This contradicts the idea behind improving network resilience and will be infeasible in real-world applications. Our goal is to enhance network resilience with moderate loss of network utility via edge rewiring.

**Graph structure learning.**    Unlike the graph generation task which focuses on the quality of the generated graph, graph structure learning (GSL) aims to jointly learn an optimized graph structure and corresponding graph representations only for better performance on downstream tasks. Although conceptually related, GSL differs from graph generation since GSL mostly cares for the downstream task while graph generation focuses on the generated graphs Jin et al. (2020); Zhu et al. (2021). Currently, GSL relies on the existence of rich features to construct a graph, while there are generally no rich features in graph generation. Moreover, GSL cannot control the graph's node degree during graph optimization. We refer the interested readers to a survey of GSL Zhu et al. (2021) since our work is a constrained graph generation task unrelated to GSL.

**Graph rewiring**    Graph rewiring is typically used in the GNN community to build novel classes of GNNs by preprocessing a given graph to overcome the problems of the over-squashing issue of training GNNs. For example, Klicpera et al. (2019) developed graph diffusion convolution (GDC) to improve GNN's performance on downstream tasks by replacing message passing with graph diffusion convolution (Topping et al., 2022) proposed an edge-based combinatorial curvature to help alleviate the over-squashing phenomenon in GNNs. To our knowledge, there is currently no inductive learning-based graph rewiring method, and graph rewiring methods rely on rich features to train GNNs better on downstream tasks. The edge rewiring operation used in our paper is a special graph rewiring operator that preserves node degree.

**Multi-views graph augmentation for GNNs.**    Multi-views graph augmentation is one efficient way to improve the expressive power of GNNs or combine domain knowledge, which is adapted based on the task's prior Hu et al. (2020). For example, GCC generates multiple subgraphs from the same ego network Qiu et al. (2020). GCA adaptively incorporates various priors for topological and semantic aspects of the graph You et al. (2020). Hassani & Khasahmadi (2020) contrasts representations from first-order neighbors and a graph diffusion. DeGNN Jin et al. (2020) was proposed as an automatic graph decomposition algorithm to improve the performance of deeper GNNs. These techniques rely on the existence of rich graph feature and the resultant GNNs cannot work well on graphs without rich features. In the resilience task, only the graph topological structure is available. Motivated by the calculation process of persistent homology Edelsbrunner & Harer (2008), we apply the filtration process to enhance the expressive power of GNNs for handling graphs without rich features.

**Deep graph generation.**    Deep graph generation models learn the distribution of given graphs and generate more novel graphs. Some work use the encoder-decoder framework by learning latent representation of the input graph through the encoder and then generating the target graph through the decoder. For example, GCPN You et al. (2018) incorporates chemistry domain rules on molecular graph generation. GT-GAN Guo et al. (2022) proposes a GAN-based model on malware cyber-network synthesis. GraphOpt Trivedi et al. (2020) learns an implicit model to discover an underlying optimization mechanism of the graph generation using inverse reinforcement learning. GFlowNet learns a stochastic policy for generating molecules with the probability proportional to a given reward based on flow networks and local flow-matching conditions Bengio et al. (2021). Boosting network resilience in a degree-preserving way can be viewed as a constrained graph generation task. However, none of existing graph generation methods can generate desired graphs with the exact node degree preserving constraint, which is required by the resilience task.

Table 3: Statistics of graphs used for resilience maximization. Both transductive and inductive settings ($\star$) are included. Consistent with our implementation, we report the number of edges by transforming undirected graphs to directed graphs. The edge rewiring has a fixed execution order. For the inductive setting, we report the maximum number of edges. The action space size of the edge rewiring is measured by $2|E|^2$.

| Dataset | Node | Edge | Action Space Size | Train/Test | Setting |
|---|---|---|---|---|---|
| BA-15 | 15 | 54 | 5832 | ✗ | Transductive |
| BA-50 | 50 | 192 | 73728 | ✗ | Transductive |
| BA-100 | 100 | 392 | 307328 | ✗ | Transductive |
| BA-500 | 500 | 996 | 1984032 | ✗ | Transductive |
| BA-1000 | 1000 | 999 | 1996002 | ✗ | Transductive |
| EU | 217 | 640 | 819200 | ✗ | Transductive |
| p2p-Gnutella05 | 400 | 814 | 1325192 | ✗ | Transductive |
| p2p-Gnutella09 | 300 | 740 | 1095200 | ✗ | Transductive |
| BA-10-30 ($\star$) | 10-30 | 112 | 25088 | 1000/500 | Inductive |
| BA-20-200 ($\star$) | 20-200 | 792 | 1254528 | 4500/360 | Inductive |

## B  Implementation Details

This section provides the implementation details, including dataset and baseline setup.

### B.1  Dataset

We first present the data generation strategies. Table 3 summarizes the statistics of each dataset. Synthetic datasets are generated using the Barabasi-Albert (BA) model (known as scale-free graphs) (Albert & Barabási, 2002), with the graph size varying from $|N|$=10 to $|N|$=1000. During the data generation process, each node is connected to two existing nodes for graphs with no more than 500 nodes, and each node is connected to one existing node for graphs with near 1000 nodes. BA graphs are chosen since they are vulnerable to malicious attacks and are commonly used to test network resilience optimization algorithms (Bollobás & Riordan, 2004). We test the performance of ResiNet on both transductive and inductive settings.

- **Transductive setting.**   The algorithm is trained and tested on the same network.

  - Randomly generated synthetic BA networks, denoted by BA-$m$, are adopted to test the performance of ResiNet on networks of various sizes, where $m \in \{15, 50, 100, 500, 1000\}$ is the graph size.

  - The Gnutella peer-to-peer network file sharing network from August 2002 (Leskovec et al., 2007; Ripeanu et al., 2002) and the real EU power network (Zhou & Bialek, 2005) are used to validate the performance of ResiNet on real networks. The random walk sampling strategy is used to derive a representative sample subgraph with hundreds of nodes from the Gnutella peer-to-peer network (Leskovec & Faloutsos, 2006).

- **Inductive setting.**   Two groups of synthetic BA networks denoted by BA-$m$-$n$ are randomly generated to test ResiNet's inductivity, where $m$ is the minimal graph size, and $n$ indicates the maximal graph size. We first randomly generate the fixed number of BA networks as the training data to train ResiNet and then evaluate ResiNet's performance directly on the test dataset without any additional optimization.

### B.2  Baseline Setup

All baselines share the same action space with ResiNet and use the same action masking strategy to block invalid actions as ResiNet does. The maximal objective evaluation is consistent for all algorithms. Other settings of baselines are consistent with the default values in their paper. The early-stopping strategy is used for baselines, which means that the search process terminates if no improvement is obtained in successive

Table 4: The effect of the filtration order $K$ on ResiNet in improving network resilience (percentage).

| $K$ | BA-15 | BA-50 | BA-100 | EU |
|---|---|---|---|---|
| 0 | $11.8 \pm 2.7$ | $0 \pm 0$ | $0 \pm 0$ | $6.1 \pm 4.2$ |
| 1 | $17.6 \pm 0$ | $49.6 \pm 2.3$ | $74.9 \pm 0.8$ | $54.5 \pm 0.4$ |
| 2 | $17.6 \pm 0$ | $51.0 \pm 0.1$ | $76.3 \pm 1.2$ | $57.4 \pm 1.6$ |
| 3 | $17.6 \pm 0$ | $55.7 \pm 2.3$ | $73.1 \pm 0.9$ | $54.9 \pm 0.9$ |

1000 objective function calling trials. All algorithms are repeated for 3 random seeds using default hyper-parameters.

## C    Extended Experimental Results

In this section, we present additional experimental results to show that ResiNet generalizes to unseen graphs, different utility and resilience metrics.

### C.1    The Effect of Filtration Order $K$ on FireGNN

In this section, we report the ratio of the remaining edges in subgraphs versus different filtration order $K$ in Table 5 and visualize it in Figure 7. The maximum filtration order $K$ of FireGNN of each dataset is summarized in Table 6. Table 6 demonstrates that the maximum filtration order $K$ is nearly around the half size of the graph since we gradually remove the node with the largest degree in the filtration process.

### C.2    Inductivity on Larger Datasets

To demonstrate that ResiNet can learn from networks to accommodate different utility and resilience metrics, we conduct experiments based on BA-15 using multiple resilience and utility metrics. The Pareto points shown in Figure 6 denote the optimum under different objectives on BA-15, implying that ResiNet can obtain the approximate Pareto frontier. Surprisingly, the initial gain of

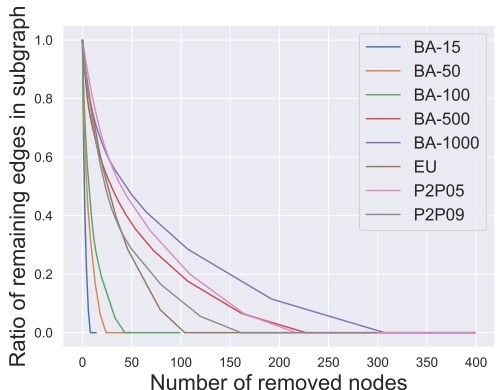

Figure 7: Ratio of the remaining edges in subgraphs versus different filtration order $K$.

resilience (from around 0.21 to around 0.24) is obtained without loss of the utility, which incentivizes almost every network to conduct such optimization to some extent when feasible. More results are included in Appendix C.3 and the optimized network structures are visualized in Figure 9 and Figure 10.

Even with limited computational resources, armed with the autoregressive action space and the power of FireGNN, ResiNet can be trained fully end-to-end on graphs with thousands of nodes using RL. We demonstrate the inductivity of ResiNet on graphs of different sizes by training ResiNet on the BA-20-200 dataset, which consists of graphs with the size ranging from 20 to 200, and then report its performance on directly guiding the edges selections on unseen test graphs. The filtration order $K$ is set to 1 for the computational limitation. As shown in Figure 8, we can see that ResiNet has the best performance for $N \in [70, 100]$. The degrading performance with the graph size may be explained by the fact that larger graphs require a larger filtration order for ResiNet to work well. A more stable performance improvement of ResiNet is observed with the increment of graph size when trained to optimize network resilience and utility simultaneously, and ResiNet possibly finds a strategy to balance these two metrics.

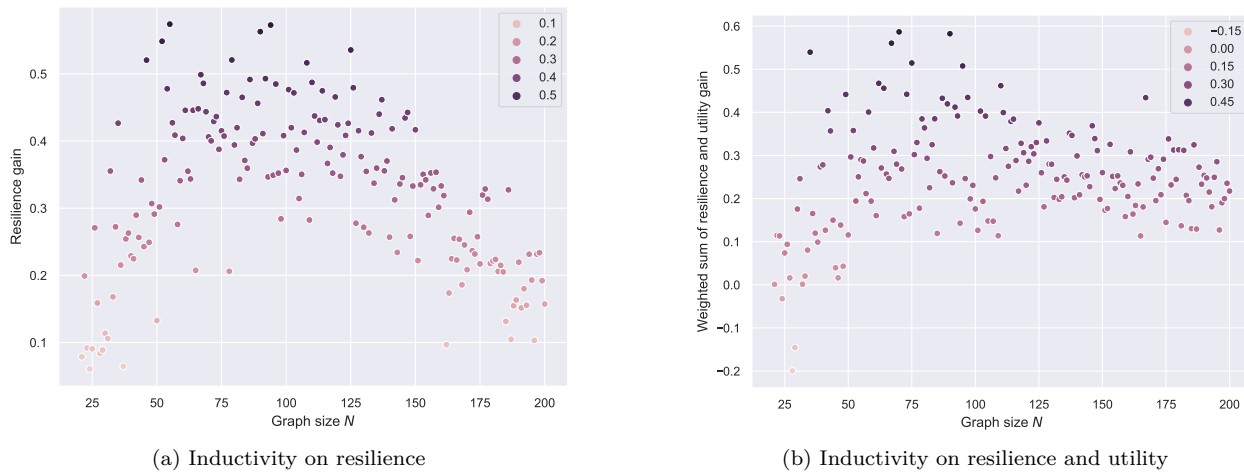

<table>
<tr><td>(a) Inductivity on resilience</td><td>(b) Inductivity on resilience and utility</td></tr>
</table>

Figure 8: The inductive ability of ResiNet on the test dataset (BA-20-200) when optimizing (a) network resilience and (b) the combination of resilience and utility.

Table 5: The ratio of remaining edges in subgraphs versus the filtration order $K$.

| $K$ | BA-15 | BA-50 | BA-100 | BA-500 | BA-1000 | EU | P2P-Gnutella05 | P2P-Gnutella09 |
|---|---|---|---|---|---|---|---|---|
| 1 | 0.6667 | 0.7708 | 0.8571 | 0.9538 | 0.9479 | 0.9625 | 0.9779 | 0.9608 |
| 2 | 0.4815 | 0.6667 | 0.7602 | 0.9096 | 0.9139 | 0.9313 | 0.957 | 0.927 |
| 3 | 0.3333 | 0.5938 | 0.6837 | 0.8665 | 0.8809 | 0.9031 | 0.9361 | 0.8973 |
| 4 | 0.2222 | 0.5208 | 0.6224 | 0.8313 | 0.8559 | 0.8781 | 0.9152 | 0.8703 |
| 5 | 0.1481 | 0.4583 | 0.5663 | 0.8002 | 0.8348 | 0.8562 | 0.8956 | 0.8446 |
| 6 | 0.0370 | 0.4062 | 0.5255 | 0.7741 | 0.8158 | 0.8344 | 0.8771 | 0.8189 |
| 7 | 0 | 0.3646 | 0.4847 | 0.754 | 0.7998 | 0.8125 | 0.8587 | 0.7959 |
| 8 | 0 | 0.3229 | 0.4439 | 0.7359 | 0.7848 | 0.7906 | 0.8403 | 0.773 |
| 9 | 0 | 0.2917 | 0.4031 | 0.7199 | 0.7698 | 0.7719 | 0.8231 | 0.75 |

Table 6: Maximum filtration order $K$ of each dataset.

| Dataset | BA-15 | BA-50 | BA-100 | BA-500 | BA-1000 | EU | P2P-Gnutella05 | P2P-Gnutella09 |
|---|---|---|---|---|---|---|---|---|
| Size (node) | 15 | 50 | 100 | 500 | 1000 | 217 | 400 | 300 |
| Maximum of K | 7 | 23 | 42 | 226 | 306 | 103 | 216 | 160 |

## C.3 Learning to Balance Different Utility and Resilience Metrics

As shown in Figure 9, we conduct extensive experiments on the BA-15 network to demonstrate that ResiNet can learn to optimize graphs with different resilience and utility metrics and to defend against other types of attacks besides the node degree-based attack, such as the node betweenness-based attack.

Table 7 records the improvements in percentage of ResiNet for varying objectives on the BA-15 dataset. As visualized in Figure 9, ResiNet is not limited to defend against the node degree-based attack (Figure 9 (b)-(j)) and also learns to defend against the betweenness-based attack (Figure 9 (k)-(s)). Total three resilience metrics are used, with $\mathcal{R}$ denoting the graph connectivity-based resilience metric, $\mathcal{SR}$ being the spectral radius and $\mathcal{SR}$ representing the algebraic connectivity. Total two utility metrics are adopted, including the global efficiency $E_{global}$ and the local efficiency $E_{local}$. Not surprisingly, the optimized network with an improvement of about 3.6% for defending the betweenness-based attack also has a higher resilience (around 7.8%) against the node-degree attack. This may be explained as the similarity between node degree and betweenness for a small network.

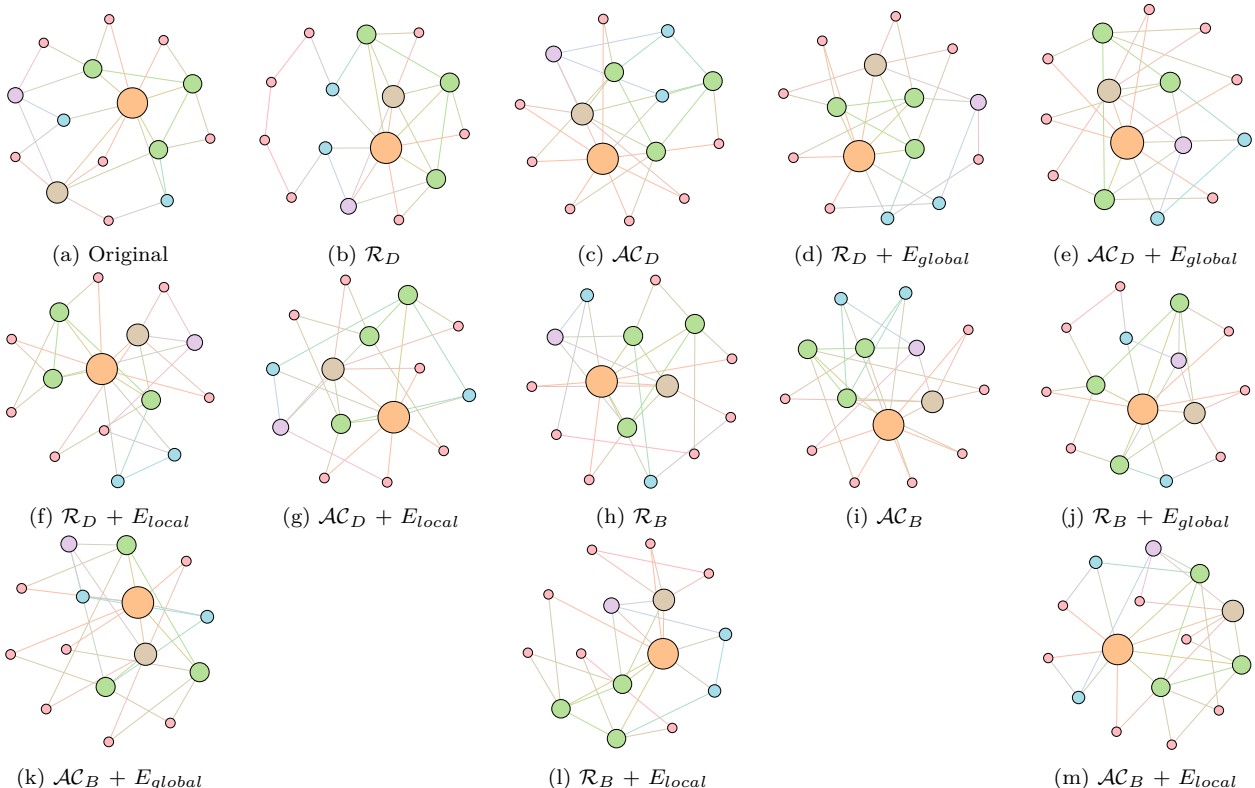

(a) Original    (b) $\mathcal{R}_D$    (c) $\mathcal{AC}_D$    (d) $\mathcal{R}_D + E_{global}$    (e) $\mathcal{AC}_D + E_{global}$

(f) $\mathcal{R}_D + E_{local}$    (g) $\mathcal{AC}_D + E_{local}$    (h) $\mathcal{R}_B$    (i) $\mathcal{AC}_B$    (j) $\mathcal{R}_B + E_{global}$

(k) $\mathcal{AC}_B + E_{global}$    (l) $\mathcal{R}_B + E_{local}$    (m) $\mathcal{AC}_B + E_{local}$

Figure 9: The resilience maximization on the BA-15 dataset with 15 nodes and 27 edges with (a) original network, (b)-(j) results of defending the node degree-based attack with different combinations of resilience and utility, and (k)-(s) results of defending against the node betweenness-based attack with varying combinations of resilience and utility. For two resilience metrics, $\mathcal{R}$ denotes the graph connectivity-based resilience metric and $\mathcal{AC}$ represents the algebraic connectivity. For two utility metrics, $E_{global}$ denotes the global efficiency and $E_{local}$ is the local efficiency.

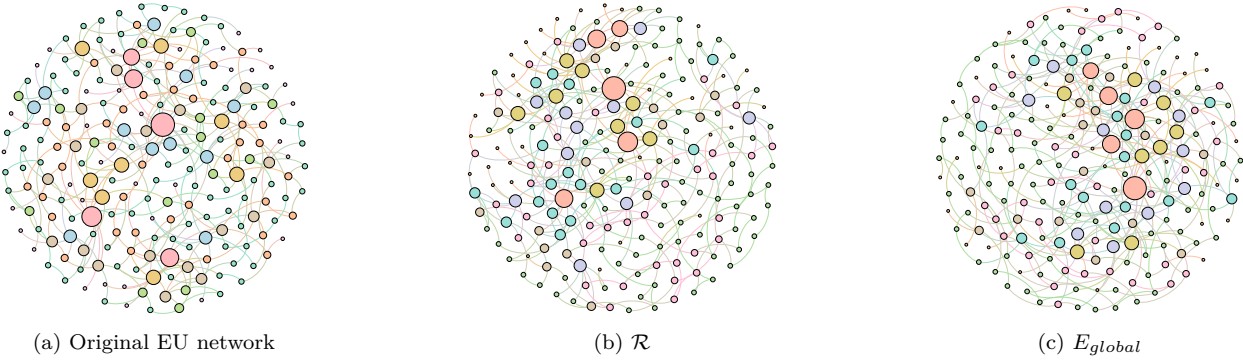

(a) Original EU network      (b) $\mathcal{R}$      (c) $E_{global}$

Figure 10: Visualizations of the original EU network and optimized networks using ResiNet with different objectives: $\mathcal{R}$ means the connectivity-based resilience measurement and $E_{global}$ is the global efficiency.

Table 7: Performance gain (in percentage) of ResiNet in optimizing varying objectives on the BA-15 network. All objectives are optimized with the same hyper-parameters, which means that we did not tune hyper-parameters for objectives except for $R_D$.

| Objective | Gain (%) | Objective | Gain(%) |
|---|---|---|---|
| $\mathcal{R}_D$ | 35.3 | $\mathcal{R}_B$ | 14.6 |
| $\mathcal{AC}_D$ | 48.2 | $\mathcal{AC}_B$ | 43.2 |

Table 8: Resilience optimization algorithm under the fixed maximal rewiring number budget of 200. Entries are in the format of $X(Y)$, where 1) $X$: weighted sum of the graph connectivity-based resilience and the network efficiency improvement (in percentage); 2) $Y$: required rewiring number. Results are averaged over 3 runs and best performance is in bold.

| Method | $\alpha$ | BA-15 | BA-50 | BA-100 | BA-500 | BA-1000 | EU | P2P-Gnutella05 | P2P-Gnutella09 |
|---|---|---|---|---|---|---|---|---|---|
| HC | 0 | 26.8 (10.0) | 52.1 (47.0) | 76.9 (97.3) | 45.8 (200) | 302.5 (200) | 71.9 (152.7) | 37.5 (193.3) | 40.2 (137.7) |
| | 0.5 | 18.6 (11.3) | 43.1 (62.7) | 56.9 (121) | 30.0 (200) | 66.3 (200) | 63.2 (200) | 27.7 (200) | 34.7 (196.3) |
| SA | 0 | 26.8 (20) | 49.7 (59.0) | 84.5 (119.7) | 43.2 (200) | 271.8 (200) | 73.5(160.3) | 37.1 (200) | 37.2 (134) |
| | 0.5 | 17.8 (21) | 41.1 (79.7) | 57.7 (127.7) | 31.4 (200) | 64.9 (200) | 62.8 (200) | 37.1 (200) | 35.2 (200) |
| Greedy | 0 | 23.5 (6) | 48.6 (13) | 64.3 (20) | ✗ | ✗ | 0.5 (3) | ✗ | ✗ |
| | 0.5 | 5.3 (15) | 34.7 (13) | 42.7 (20) | ✗ | ✗ | 0.3 (3) | ✗ | ✗ |
| EA | 0 | 35.3 (✗) | 50.2 (✗) | 61.9 (✗) | 9.9 (200) | 174.1 (200) | 66.2 (✗) | 2.3 (200) | 0 (200) |
| | 0.5 | 27.1 (✗) | 38.3 (✗) | 46.6 (✗) | 6.8 (200) | 18.7 (200) | 58.4 (✗) | 3.2 (200) | 0 (200) |
| DE-GNN-RL | 0 | 13.7 (2) | 0 (1) | 0 (1) | 1.6 (20) | 41.7 (20) | 9.0 (20) | 2.2 (20) | 0 (1) |
| | 0.5 | 10.9 (2) | 0 (1) | 0 (1) | 2.7 (20) | 20.1 (14) | 2.1 (20) | 0 (1) | 1.0 (20) |
| $k$-GNN-RL | 0 | 13.7 (2) | 0 (1) | 0 (1) | 0 (1) | 8.8 (20) | 4.5 (20) | -0.2 (20) | 0 (1) |
| | 0.5 | 6.3 (2) | 0 (1) | 0 (1) | 0 (20) | -24.9 (20) | 4.8 (20) | -0.1 (20) | 0 (1) |
| ResiNet | 0 | **35.3** (6) | **61.5** (20) | **70.0** (20) | **10.2** (20) | 172.8 (20) | **54.2** (20) | **14.0** (20) | **18.6** (20) |
| | 0.5 | **26.9** (20) | **53.9** (20) | **53.1** (20) | **15.7** (20) | **43.7** (20) | **51.8** (20) | **12.4** (20) | **15.1** (20) |

## C.4 Performance Comparisons Under a Large Rewiring Budget

In this section, we present the resilience improvement and the required number of edge rewiring of each algorithm under a large rewiring budget of 200. The running speed is presented to compare the running time efficiency of each algorithm.

As shown in Table 8, traditional methods improve the network resilience significantly compared to ResiNet under a large rewiring budget of 200. However, traditional methods are still undesired in such a case since a solution with a large rewiring budget is not applicable in practice due to the vast cost of adding many new edges into a real system. For example, the actual number of rewiring budget for EA is hard to calculate since it is a population-based algorithm, so it is omitted in Table 8. All baselines adopt the early-stopping strategy that they will terminate if there is no positive resilience gain in a successive 1000 steps.

Table 9 indicates that the time it takes for the benchmark algorithm to solve the problem usually increases as the test data set size increases. In contrast, our proposed ResiNet is suitable for testing on a large dataset once trained.

Table 9: Running speed (in second) of the resilience optimization algorithm under the fixed maximal rewiring number budget. Entries are in the format of $X(Y)$, where 1) $X$: speed under the budget of 20; 2) $Y$: speed under the budget of 200 . ✗ means that the result is not available at a reasonable time. Results are averaged over 3 runs and best performance is in bold.

| Method | $\alpha$ | BA-15 | BA-50 | BA-100 | BA-500 | BA-1000 | EU | P2P-Gnutella05 | P2P-Gnutella09 |
|---|---|---|---|---|---|---|---|---|---|
| HC | 0 | 1.0 (1.0) | 1.1 (6.4) | 1.3 (22.2) | 21.9 (354.1) | 80.3 (1288.3) | 3.1 (94.2) | 15.3 (358.1) | 4.5 (89.1) |
| | 0.5 | 1.5 (11.5) | 1.1 (12.8) | 2.0 (49.0) | 40.9 (589.5) | 148.7 (2603.7) | 5.3 (193.7) | 24.7 (462.8) | 7.0 (190.6) |
| SA | 0 | 0.5 (0.5) | 0.3 (6.6) | 0.6 (22.6) | 12.2 (313.0) | 45.7 (1051.8) | 2.4 (91.2) | 10.8 (286.4) | 2.6 (89.4) |
| | 0.5 | 0.7 (1.7) | 0.7 (13.2) | 1.7 (47.5) | 33.9 (568.9) | 99.8 (2166.3) | 5.0 (193.5) | 23.9 (454.5) | 6.3 (188.5) |
| Greedy | 0 | 0.2 (6.0) | 34.1 (34.5) | 766.3 (✗) | ✗ | ✗ | 3061.7 (✗) | ✗ | ✗ |
| | 0.5 | 0.7 (0.7) | 64.1 (65.4) | 1478.9 (✗) | ✗ | ✗ | 6192.6 (✗) | ✗ | ✗ |
| EA | 0 | 0.01 (✗) | 0.1 (✗) | 1.6 (✗) | 2.5 (✗) | 10.3 (✗) | 0.2 (✗) | 1.6 (✗) | 0.4 (✗) |
| | 0.5 | 0.01 (✗) | 0.1 (✗) | 0.8 (✗) | 4.7 (✗) | 15.0 (✗) | 0.4 (✗) | 3.0 (✗) | 0.8 (✗) |
| DE-GNN-RL | 0 | 0.1 (✗) | 0.1 (✗) | 0.1 (✗) | 14.9 (✗) | 70.3 (✗) | 3.6 (✗) | 8.7 (✗) | 0.5 (✗) |
| | 0.5 | 0.1 (✗) | 0.1 (✗) | 0.2 (✗) | 13.7 (✗) | 60.9 (✗) | 4.5 (✗) | 1.0 (✗) | 6.7 (✗) |
| $k$-GNN-RL | 0 | 0.02 (✗) | 0.03 (✗) | 0.07 (✗) | 1.3 (✗) | 56.5 (✗) | 2.6 (✗) | 8.2 (✗) | 0.5 (✗) |
| | 0.5 | 0.02 (✗) | 0.04 (✗) | 0.08 (✗) | 18.3 (✗) | 76.1 (✗) | 3.6 (✗) | 11.5 (✗) | 0.6 (✗) |
| ResiNet | 0 | 0.5 (✗) | 1.8 (✗) | 2.2 (✗) | 17.5 (✗) | 66.8 (✗) | 4.5 (✗) | 14.7 (✗) | 9.3 (✗) |
| | 0.5 | 0.5 (✗) | 1.9 (✗) | 2.4 (✗) | 18.0 (✗) | 67.5 (✗) | 5.2 (✗) | 15.0 (✗) | 10.3 (✗) |

### C.5 Inspection of Optimized Networks

Moreover, to provide a deeper inspection into the optimized network structure, we take the EU power network as an example to visualize its network structure and the optimized networks given by ResiNet with different objectives. Compared to the original EU network, Figure 10 (b) is the network structure obtained by only optimizing the graph connectivity-based resilience. We can observe a more crowded region on the left, consistent with the "onion-like" structure concluded in previous studies. If we consider the combination gain of both resilience and utility, we observe a more compact clustering "crescent moon"-like structure as shown in Figure 10 (c).

## D Deep analysis of why regular GNNs fail in the resilience task

It is well-known that GNNs generally work well for graphs with rich features. Unluckily, the graph network in the resilience task has no node/edge/graph feature, with only the topological structure available. No rich feature means that the output of the GNNs is not distinguishable, and then it is difficult for the RL agent to distinguish different vertices/edges, causing large randomness in the output of the policy. This may cause the rewiring process to alternate between two graphs, forming an *infinite loop*. And we suspect that this infinite loop failure may explain the poor empirical performance of optimizing network resilience by selecting edges using existing GNNs and reinforcement learning (RL). The infinite loop failure is presented as follows.

Consider the graph $G_t$ with $N$ nodes and containing two edges $AB$ and $CD$. The agent selects $AB$ and $CD$ for rewiring, leading to $G_{t+1}$ with news edges $AC$ and $BD$. A frequent empirical failure of regular GNNs for the robustness task is the *infinite action backtracking* phenomenon. The agent would select $AC$ and $BD$ at step $t + 1$, returning back to $G_t$ and forming a cycled loop between $G_t$ and $G_{t+1}$. Formally, the infinite loop is formulated as

$$((A, B), (C, D)) = \operatorname*{argmax}_{i,j,m,n \in 1:N} \operatorname{SIM}\left(((h_t^i, h_t^j), (h_t^m, h_t^n)), h_{G_t}\right)$$

$$((A, C), (B, D)) = \operatorname*{argmax}_{i,j,m,n \in 1:N} \operatorname{SIM}\left(((h_{t+1}^i, h_{t+1}^j), (h_{t+1}^m, h_{t+1}^n)), h_{G_{t+1}}\right) ,$$

where SIM is a similarity metric, $h_t^i$ and $h_{G_t}$ are embeddings of node $i$ and graph $G_t$ at step $t$, and $(A,B)$ is one edge.

Table 10 compares and summarizes different graph related tasks' characteristics. We can see that *the resilience task is more challenging from many aspects*. No prior rule like action masking or negative penalty can be used to avoid selecting visited nodes as in TSP. For the resilience task, all previously visited edges are also possibly valid to be selected again, resulting in insufficient training signals.

The desired GNN model should not depend on rules like action masking to distinguish edge and graph representations for graphs with little node features. Our proposed FireGNN fulfills these requirements to obtain proper training signals. FireGNN has a distinct expressive power and learns to create more meaningful and distinguishable features for each edge. FireGNN is not a simple aggregation of higher-order information of a static graph. It was inspired by homology filtration and the multi-view graph augmentation. Persistence homology motivates us to aggregate more distinct node features by observing how the graph evolves towards being empty, leading to more distinct and meaningful features for each node/edge, thus avoiding the infinite loop. Extensive experimental results in Table 1 validate the necessity and effectiveness of FireGNN. Existing GNNs perform worse while FireGNN performs well.

Table 10: Characteristics of different graph related tasks.

| Approach | Task | RL component | | | Problem Complexity | Size Extrapolate | Training & Inference | | |
|---|---|---|---|---|---|---|---|---|---|
| | | State | Action | Reward | | | Encoder | Action Masking | Scalability |
| S2V-DQN Khalil et al. (2017) | MVC | node level | add node to subset | -1 | $\mathcal{O}(N)$ | ✗ | S2V | ✓ | 500 |
| | Max-Cut | node level | add node to subset | change in cut weight | $\mathcal{O}(N)$ | ✗ | S2V | ✓ | 300 |
| | TSP | node level | add node to tour | change in tour cost | $\mathcal{O}(N)$ | ✗ | S2V | ✓ | 300 |
| Local search Hudson et al. (2022) | TSP | edge level | relocate node in tour | global regret | $\mathcal{O}(N^2)$ | ✓ | GNN | ✓ | 100 |
| RNN-RL Bello et al. (2016) | TSP | node level | add node to tour | change in tour cost | $\mathcal{O}(N)$ | ✗ | RNN | ✓ | 100 |
| GNN-RL Joshi et al. (2022) | TSP | node level | add node to tour | change in tour cost | $\mathcal{O}(N)$ | ✓ | GNN | ✓ | 50 |
| Attention-RL Kool et al. (2018) | TSP | node level | add node to tour | change in tour cost | $\mathcal{O}(N)$ | ✗ | Attention | ✓ | 100 |
| | VRP | node level | add node to tour | change in tour cost | $\mathcal{O}(N)$ | ✗ | Attention | ✓ | 100 |
| Local search Böther et al. (2022) | MIS | node level | add node to subset | change in IS size | $\mathcal{O}(N)$ | ✓ | GNN | ✓ | 800 |
| RNet-DQN Darvariu et al. (2021) | Resilience | graph level | edge addition | change in resilience (only at the end of the episode) | $\mathcal{O}(N^2)$ | ✓ | GNN | ✓ | 20 |
| SG-UCT Darvariu et al. (2023) | Resilience/efficiency | graph level | edge addition | change in resilience/efficiency (only at the end of the episode) | $\mathcal{O}(N^2)$ | ✓ | GNN | ✓ | 150 |
| **ResiNet** | Resilience | *graph level* | *edge rewiring* | change in resilience and utility | $\mathcal{O}(N^4)$ | ✓ | *FireGNN* | ✗ | *1000* |

