# OpenReview forum: "Learning to Boost Resilience of Complex Networks via Neural Edge Rewiring"
_TMLR — Accepted by TMLR_

### Review · Reviewer_PxTN · 2023-06-17

**Summary Of Contributions:**

This submission presents a new algorithm for rewiring networks under the
constraints of improving resilience and maintaining network utility. The
new method is inspired by concepts from computational topology, and uses
such concepts to learn structure-based node representations. Following a
representation learning step, network structure is optimised by treating
the rewiring operation as a Markov decision process.


**Audience:**

Yes

**Broader Impact Concerns:**

There are no broader impact concerns to be raised by this work.


**Claims And Evidence:**

Yes

**Requested Changes:**

- Please update the rewiring example in the introduction and make it
  clear *why* some of the edges must not exist (i.e. in order to still
  have the degree-preserving property).

- Please provide more details on the filtration and its definition. In
  computational topology, filtrations are typically used to assess
  functions on structured data via multiple scales. Can you link this
  motivation to your work?

- When talking about a 'temporal-related filtration,' this might be
  misleading since the time aspect is referring to the algorithm itself
  rather than some time encoded on the graph edges; I would suggest to
  clear that up.

- The terminology for resilience and utility is not used consistently in
  Section 3.

- Please highlight the relevant edges in Figure 4, I found this figure
  rather confusing at my first read of the paper.

- When discussing parts of the method in Section 4.1, please provide
  more background on RL. Moreover, please address to what extent
  'tottering' or 'loops' are prevented; is it possible that the method
  will flip back and forth between a set of edges, or is this prevented
  by your method?

- How is $K$ picked in practice? Please add more details.

- What are the reasons for choosing the pointer network? Please clarify.

- When discussing the experiments, please clarify to what extent other
  methods can be trained with a similar reward. My impression was that
  the new objective or reward was designed specifically for the problem
  at hand. Is it easy to add to existing methods or does the $\alpha$
  parameter only refer to a post-hoc evaluation (as opposed to a full
  training)?

- Please comment on the **stability** and **limitations** of the
  proposed approach. Some of the existing rewiring algorithms, e.g. ones
  based on the `PageRank` algorithm or on diffusion, come equipped with
  stability guarantees and a well-understood behaviour under
  perturbations. What types of guarantees hold for the proposed method?
  Are there situations in which the method does not work as expected?

Some minor points concerning the wording and typesetting:

- The main contributions of the paper are duplicated on p. 3

- 'pure topology structure' --> 'pure topological structure'

- 'the environment performs [...]': environment does not seem to be the
  correct word here.

- Please perform another round of consistency checks for the
  bibliography. At present, some entries are not consistently spelled or
  capitalised (e.g. 'lehman' instead of 'Lehman').


**Strengths And Weaknesses:**

The main strengths of the paper are:

1. A principled way of addressing rewiring in a 'learning' scenario.
2. A well-principled and justified method for balancing network utility and efficiency.

At the same time, there are some weaknesses that need to be addressed
before this submission is ready for publication. The primary weakness
is a lack of clarity: some concepts (especially for RL) are lacking a
detailed description, making it harder to understand this submission,
or necessitating referring to additional literature. In addition, the
terminology is not fully consistent and some experiments require more
details.

Subsequently, I comment on some changes to rectify these issues.

---

> ### Author Response · Authors · 2023-07-03
> **Response to Reviewer PxTN (Part 1)**
>
> Thank you for your valuable comments and suggestions, which have greatly contributed to the improvement of our paper. We have marked the revisions in blue for easy identification. Please find our detailed responses to your comments below.
>
> > Please update the rewiring example in the introduction and make it clear why some of the edges must not exist (i.e. in order to still have the degree-preserving property).
>
>    In the revised version, we have updated the rewiring example in the introduction to clarify the necessity of certain edges not existing in order to maintain the degree-preserving property.
>
> > Please provide more details on the filtration and its definition. In computational topology, filtrations are typically used to assess functions on structured data via multiple scales. Can you link this motivation to your work?
>
> We have provided more details and the definition of the filtration in the introduction and linked it to our work as a motivation.
>
> > When talking about a 'temporal-related filtration,' this might be misleading since the time aspect is referring to the algorithm itself rather than some time encoded on the graph edges; I would suggest to clear that up.
>
> In order to avoid possible misunderstanding, we have replaced the term "temporal" with "sequential" throughout the paper.
>
> > The terminology for resilience and utility is not used consistently in Section 3.
>
> We have carefully checked Section 3 to ensure the consistent use of terminology for resilience and utility.
>
> > Please highlight the relevant edges in Figure 4, I found this figure rather confusing at my first read of the paper.
>
> In  Sec 4.1 when defining the “Action” module （Page 6), we have provided a detailed description of the edge rewiring  action space. As a result, we found that Figure 4 became redundant in illustrating the same concept. Considering the limited space available, we have decided to remove Figure 4 from the revised version of the paper.
>
> > When discussing parts of the method in Section 4.1, please provide more background on RL.
>
> We have provided more background on RL in Section 4.1.
>
> > Moreover, please address to what extent 'tottering' or 'loops' are prevented; is it possible that the method will flip back and forth between a set of edges, or is this prevented by your method?
>
> In the revised introduction (Page 3), we have made it clearer that our method does not suffer from the issue of “loops”  by emphasizing that our proposed FireGNN can provide distinguishable edge embeddings. This allows the RL agent to effectively learn the selection of edges without falling into undesired infinite action backtracking loops.
>
> > How is $K$ picked in practice? Please add more details.
>
> We have discussed practical suggestions for selecting the value of $K$ in Section 5.3 (Page 11) of the manuscript. In practice, the choice of $K$ depends on the size and complexity of the graph being considered. For small to moderate-sized graphs, it is often suitable to set $K$ to the maximum node size, as this allows for a comprehensive exploration of the filtration process. However, for larger graphs where computational limitations arise, it may be necessary to set $K$ to a smaller value such as 1 or 2.
>
> Additionally, factors such as the available computational resources and the specific requirements of the application may influence the selection of $K$. It is important to strike a balance between the computational efficiency and the desired level of resilience analysis. Researchers and practitioners can experiment with different values of $K$ to find an appropriate setting for their specific graph resilience optimization tasks.
>
> Please refer to Section 5.3 (Page 11) for more detailed discussions and practical recommendations regarding the selection of $K$ in practice.
>
> > What are the reasons for choosing the pointer network? Please clarify.
>
> In Section 4.3 (Page 9) of the manuscript, we have provided a detailed discussion on the reasons for choosing the pointer network for our resilience task. The pointer network offers several advantages that align well with the requirements of our task. The pointer network offers the advantages of handling variable-sized outputs, flexible action selection, and end-to-end training.
>
> > When discussing the experiments, please clarify to what extent other methods can be trained with a similar reward. My impression was that the new objective or reward was designed specifically for the problem at hand. Is it easy to add to existing methods or does the $\alpha$ parameter only refer to a post-hoc evaluation (as opposed to a full training)?
>
> In Section 5.1 (Page 9), we have made it clearer that all baselines are trained using the same reward as ResiNet. To incorporate a new objective function or reward function into existing methods, we follow a similar process as we did with ResiNet by evaluating the new objective or reward with different $\alpha$.

---

> > ### Comment · Reviewer_PxTN · 2023-07-04
> >
> > Thanks for this extensive rebuttal and your updates! I have only spotted one minor issue: in 'Limitations and Future Work,' please use _persistent homology_ instead of _filtration homology_. Other than that, my queries have been answered and all my concerns have been addressed.

---

> > > ### Author Response · Authors · 2023-07-04
> > > **Thank you**
> > >
> > > Dear Reviewer,
> > >
> > > Thank you for your feedback on our paper. We have carefully revised the manuscript and addressed the issue raised in the 'Limitations and Future Work' section. Specifically, we have changed *filtration homology* to *persistent homology*.
> > >
> > > Once again, we sincerely appreciate your valuable comments and suggestions, which have greatly contributed to the improvement of our work.
> > >
> > > Best regards,
> > >
> > > The Authors

---

> ### Author Response · Authors · 2023-07-03
> **Response to Reviewer PxTN (Part 2)**
>
> > Please comment on the stability and limitations of the proposed approach. Some of the existing rewiring algorithms, e.g. ones based on the PageRank algorithm or on diffusion, come equipped with stability guarantees and a well-understood behavior under perturbations. What types of guarantees hold for the proposed method? Are there situations in which the method does not work as expected?
>
> In the revised version of the paper, we have included a stability analysis in Section 5.4 and a limitation analysis in Section 5.5 to provide a more comprehensive understanding of our proposed method.
>
> > Some minor points concerning the wording and typesetting:
>
> We have carefully revised our paper  to address concerns related to wording and typesetting.

---

### Review · Reviewer_NdKT · 2023-06-21

**Summary Of Contributions:**

This paper investigates the resilience bossting problem in complex networks via edge rewiring. Specifically, existing non-learning based methods have limited generalization capability, while GNNs cannot be straightforwardly applied to resolve this problem due to the lack of rich node attributes in the input data.

The soluton provided in this paper consists of a FireGNN that generates node representations solely based on the graph structures, and a ResiNet that models the resilience bossting as a Markov decision process via the edge rewiring actions.


**Audience:**

Yes

**Broader Impact Concerns:**

Complex networks are pervasive in our world, therefore this work is related to multiple perspectives like electricity network, the Internet, etc.

**Claims And Evidence:**

No

**Requested Changes:**

It would be better if the concerns above are addressed.

**Strengths And Weaknesses:**

Strengths

This paper studies a meaningful problem to boost the resilience of complex networks, which are pervasive in our world. The proposed method is a novel learning based method, and the experiments are conducted comprehensively on multiple datasets.

Weaknesses

Some parts are not clear enough, e.g. the motivation of the FireGNN design is unclear, which is a major concern of this work, please see the concerns below:

1. Edge rewiring protects the capacity constraint, while edge addition or deletion cannot. But what if the capacity constraints are considered and the edge addtion is required to not exceed the constraint? Is there any advantage of edge addition and deletion over rewiring?

2.  Edge rewiring minimizes the utility degradation in terms of graph Lapalacian. Why is this important? Is this only important to specific applications like electricity networks?

3. It is stated that learning-free methods has poor transduction capability. However, since the learning-free methods do not learn anything from the graph, when a new graph is given, the algorithm still performs the same procedure, why is this concerened with transduction problem? Learning based methods may experience overfitting and lack of transduction/generalization capability. They also need training process causing extra cost. But the learning-free methods do not sound like to have these issues.

4. With GNNs, the node attributes can also be initialized as the degrees or other graph statistics, this strategies are widely accepted for graphs without node attributes, therefore it is not clear why cannot GNNs be directly used.

5.  Why is the FireGNN designed to operate on a sequence of graphs obtained by filtraton process? Why is the concatenation of the node representation obtained at different graph snapshots beneficial for the target task. How is persistent homology related to this work? Also, it is unclear how is the input  node attribute initialized.

6. Why are the resilience metrics able to reflect the resilience of the networks in real-world? It seems that different real-world networks function differently, and why is the resilience metrics universily applicable?

---

> ### Author Response · Authors · 2023-07-03
> **Response to Reviewer NdKT (Part 1)**
>
> We sincerely appreciate your valuable comments and suggestions, as they have been immensely helpful in enhancing the quality of our paper. We have incorporated each of your recommendations into the revised version. In the following sections, we have provided a detailed point-by-point response to address your comments.
>
> > Edge rewiring protects the capacity constraint, while edge addition or deletion cannot. But what if the capacity constraints are considered and the edge addition is required to not exceed the constraint? Is there any advantage of edge addition and deletion over rewiring?
>
>  In the revised paper (page 1), we have provided a clearer explanation to highlight the advantages and disadvantages of edge addition and deletion compared to rewiring. One advantage of edge addition is that it can often lead to better performance compared to rewiring, but this may come at the cost of changing the degree distribution. Degree is an important property of complex networks, and the conventional setting is to use edge rewiring to enhance network resilience while preserving the degree distribution. However, if capacity constraints allow for new additions without exceeding the constraint, edge addition can be a viable option.
> It is worth noting that our proposed method can easily be extended to incorporate edge additions, as the rewiring process we consider already involves edge additions.
>
> > Edge rewiring minimizes the utility degradation in terms of graph Lapalacian. Why is this important? Is this only important to specific applications like electricity networks?
>
>  In the revised Introduction, we have clarified that minimizing the utility degradation in terms of graph Laplacian is important because many important graph properties are defined based on the eigenvalues of the Laplacian matrix. The Laplacian matrix captures essential structural information about a graph, and its eigenvalues are associated with properties such as connectivity, robustness, and spectral clustering.
> Therefore, minimizing the utility degradation in terms of graph Laplacian is not limited to specific applications like electricity networks but is relevant in various domains where the utility and resilience of the underlying graph structure are crucial, including transportation networks, social networks, biological networks, and communication networks.
>
> > It is stated that learning-free methods has poor transduction capability. However, since the learning-free methods do not learn anything from the graph, when a new graph is given, the algorithm still performs the same procedure, why is this concerened with transduction problem? Learning based methods may experience overfitting and lack of transduction/generalization capability. They also need training process causing extra cost. But the learning-free methods do not sound like to have these issues.
>
> While it is true that learning-free methods perform the same procedure for a new graph, this can become time-consuming for graphs of moderate sizes. The calculation of resilience is often computationally intensive, and searching for solutions from scratch repeatedly can be inefficient. In contrast, a learning-based method, once trained, can efficiently provide solutions for new graphs without starting the search from scratch each time.
>
> Some learning-free methods rely on expert rules to guide the decision-making process. This dependence on expert knowledge can limit their applicability and flexibility. In contrast, learning-based methods can autonomously learn from data and adapt to different graph structures without the need for explicit expert rules.
>
> While the training process of learning-based methods incurs an additional cost, the offline testing on new graphs is fast since the resilience calculation is not required. Furthermore, learning-based methods have the potential for better generalization capability, as they can learn patterns and relationships from training data, allowing them to make predictions or decisions on unseen graphs based on the learned knowledge. This ability to generalize beyond the training data is a valuable characteristic in many applications.

---

> > ### Author Response · Authors · 2023-07-03
> > **Response to Reviewer NdKT (Part 2)**
> >
> > > With GNNs, the node attributes can also be initialized as the degrees or other graph statistics, this strategies are widely accepted for graphs without node attributes, therefore it is not clear why cannot GNNs be directly used.
> >
> > While it is true that node attributes can be initialized as degrees or other graph statistics, in the case of network resilience optimization, the widely-used node degree feature does not significantly benefit the task due to the degree-preserving rewiring. In this context, the goal is to optimize the resilience of a single graph by rewiring its edges while preserving the node degrees. This constraint limits the usefulness of traditional node attributes, such as degrees, as they remain unchanged during the rewiring process. We have provided a detailed analysis in Appendix D explaining why regular GNNs fail to effectively address the resilience task. Based on this analysis, it is clear that directly applying GNNs with node attributes may not provide substantial advantages for the specific task of network resilience optimization.
> >
> > > Why is the FireGNN designed to operate on a sequence of graphs obtained by filtraton process? Why is the concatenation of the node representation obtained at different graph snapshots beneficial for the target task. How is persistent homology related to this work? Also, it is unclear how is the input node attribute initialized.
> >
> >  In the revised paper, we have provided more details about persistent homology in the Introduction and its connection to our work. Persistent homology is a mathematical framework that measures the duration of specific topological properties within a simplicial complex as simplices are added or removed. The sequence of subcomplexes constructed during this process, known as filtration, captures valuable information about the resilience quality of the network. Therefore, FireGNN is designed to operate on the sequence of graphs obtained through the filtration process.
> >
> > The concatenation of node representations obtained at different graph snapshots is beneficial for the target task because it captures the evolving neighborhood structures of nodes across different subgraphs. By aggregating these representations, we can obtain a final edge embedding that is distinguishable and informative for the reinforcement learning agent to make accurate edge selection decisions. This improves the performance of FireGNN compared to regular GNNs, which struggle to provide distinguishable edge embeddings for effective selection.
> >
> > Regarding the initialization of input node attributes, we have introduced the distance encoding strategy and the 8-dimensional position embedding originating from the Transformer in Section 4.1 when defining the "State". These strategies provide initial information about the node positions and distances within the graph, which can help in capturing structural information and enhancing the learning process of FireGNN.  All learning-based baseline methods also use the same initialization of input node attributes.
> >
> > > Why are the resilience metrics able to reflect the resilience of the networks in real-world? It seems that different real-world networks function differently, and why is the resilience metrics universily applicable?
> >
> >  In our paper, we employ widely used and standard resilience metrics as representative measures to evaluate the resilience of networks. While it is true that different real-world networks can function differently, the resilience metrics themselves are designed to capture and quantify the properties and behaviors of the networks.
> >
> > Resilience metrics are typically defined based on specific characteristics and performance criteria of the networks, such as connectivity, efficiency, robustness, or the ability to withstand failures or attacks. These metrics are derived from mathematical formulations or empirical observations and provide a quantitative measure of the network's resilience. By evaluating networks using these metrics, we can gain insights into their resilience properties and compare their relative performance.
> >
> > It is important to note that resilience metrics are designed to be universally applicable and can be used to assess networks across different domains and applications. While the specific values or thresholds of the metrics may vary depending on the context, the underlying principles and concepts remain consistent. Therefore, even though real-world networks may exhibit diverse functionalities, the resilience metrics offer a standardized framework for assessing and comparing their resilience characteristics.

---

### Review · Reviewer_7uvs · 2023-06-24

**Summary Of Contributions:**

This paper introduces a novel technique to enhance the resilience of complex networks in the face of structural attacks, by employing a method called degree-preserving edge rewiring. Most existing methods do not effectively generalize across different graphs. To address this, the authors propose a new variant of Graph Neural Networks (GNNs), named Filtration enhanced GNN (FireGNN), and an inductive method called ResiNet. The FireGNN is designed to learn meaningful node representations purely from graph structures. ResiNet takes a step further by transforming the optimization of network resilience into a sequential decision-making process under the frame of a Markov decision process. Experimental results showed that ResiNet significantly outperformed existing methods, achieving near-optimal resilience on various graph structures while also preserving network utility.

**Audience:**

Yes

**Broader Impact Concerns:**

I have no concern.

**Claims And Evidence:**

Yes

**Requested Changes:**

The Strengths And Weaknesses section provides some suggestions and questions, which might help improve the quality of the paper.

**Strengths And Weaknesses:**

Strengths:
1. Innovation: The paper presents a novel approach (ResiNet and FireGNN) for improving network resilience, providing a significant advancement over existing methods.
2. Broad Applicability: The proposed methods do not require rich node features, making them widely applicable to various network systems with different graph structures.
3. Effectiveness: The experimental results provide evidence for the effectiveness of ResiNet, as it outperforms existing methods in enhancing network resilience while maintaining network utility.

Weaknesses:
1. Lack of Real-world Testing: Although the paper presents a promising approach with extensive theoretical and experimental support, it lacks a crucial element - the application of this approach in real-world network systems. While simulated environments or abstract graphs are suitable for initial testing and concept proofing, they may not encapsulate the complexity and unpredictable nature of real-world networks, which can include fluctuating network loads, varying degrees of node failures, and adaptive attacks. Without tests in practical scenarios, the utility and applicability of the proposed approach may be limited. It would be insightful to include results from tests in real-world conditions, such as telecommunication networks, power grids, or transportation systems, to demonstrate the robustness and applicability of the proposed ResiNet and FireGNN methods in these challenging environments.
2. Computational Efficiency: Another potential drawback in the paper lies in the computational efficiency of the proposed method. The optimization process described, which includes sequentially selecting the appropriate edges to rewire, could potentially be computationally expensive, especially for large-scale networks with hundreds or thousands of nodes. This can limit the method's scalability and practical utility. For instance, in a real-world scenario like an infrastructure system or a supply chain network, the method may not be feasible if it requires extensive computation resources or takes an unreasonably long time to improve the network's resilience. The paper would be strengthened by including a thorough analysis of the computational complexity of the proposed method and discussions on its scalability to larger networks.
3. Unclear Failure Scenarios: The paper presented the proposed methods in a very positive light, showing they outperform existing methods significantly. However, no method is foolproof, and it is essential to understand the limitations and potential failure points of the proposed approach. The absence of a discussion on potential limitations or failure scenarios might create a biased perspective towards the method's overall robustness. Such a discussion is important because it helps users and researchers understand when it might be unsuitable to employ the proposed methods, what issues they may encounter, and what improvements could be made in the future. For instance, how would the method perform in networks with a specific structure that makes edge rewiring difficult or in networks experiencing dynamic changes? Are there certain types of networks where the performance of the method might decline significantly? Including such discussions would provide a more balanced and comprehensive view of the proposed methods.

---

> ### Author Response · Authors · 2023-07-03
> **Response to Reviewer 7uvs (Part 1)**
>
> Dear Reviewer, thank you for your insightful review. We provide more detailed answers to your questions.
>
> > Lack of Real-world Testing: Although the paper presents a promising approach with extensive theoretical and experimental support, it lacks a crucial element - the application of this approach in real-world network systems. While simulated environments or abstract graphs are suitable for initial testing and concept proofing, they may not encapsulate the complexity and unpredictable nature of real-world networks, which can include fluctuating network loads, varying degrees of node failures, and adaptive attacks. Without tests in practical scenarios, the utility and applicability of the proposed approach may be limited. It would be insightful to include results from tests in real-world conditions, such as telecommunication networks, power grids, or transportation systems, to demonstrate the robustness and applicability of the proposed ResiNet and FireGNN methods in these challenging environments.
>
> It is worth noting that many real-world systems can be effectively abstracted and studied as complex networks. This approach is widely adopted in the field of network science.
> In our paper, we have conducted extensive evaluations and experiments on both synthetic and **real-world** datasets using established benchmarks and state-of-the-art resilience optimization baselines. These experiments follow the conventional experimental setup and allow us to rigorously assess the performance and effectiveness of our proposed ResiNet and FireGNN methods.
>
> While testing our methods on highly complex and specific real-world systems that may not be directly modeled as complex networks could provide additional insights, it is important to note that our paper's main focus is on network resilience optimization using complex network models. We believe that demonstrating the effectiveness and superiority of our proposed methods within the scope of complex network models is a significant contribution.
>
> However, we acknowledge the value of real-world testing instead of abstract graphs and its implications for practical applications. Future work could explore the application of our methods on specific real-world network systems, such as telecommunication networks, power grids, or transportation systems, to evaluate their robustness and applicability in such challenging environments.

---

> ### Author Response · Authors · 2023-07-03
> **Response to Reviewer 7uvs (Part 2)**
>
> > Computational Efficiency: Another potential drawback in the paper lies in the computational efficiency of the proposed method. The optimization process described, which includes sequentially selecting the appropriate edges to rewire, could potentially be computationally expensive, especially for large-scale networks with hundreds or thousands of nodes. This can limit the method's scalability and practical utility. For instance, in a real-world scenario like an infrastructure system or a supply chain network, the method may not be feasible if it requires extensive computation resources or takes an unreasonably long time to improve the network's resilience. The paper would be strengthened by including a thorough analysis of the computational complexity of the proposed method and discussions on its scalability to larger networks.
>
>  We appreciate your concern regarding the computational efficiency of the proposed method. In our paper, we have performed experiments on large-scale networks with hundreds or thousands of nodes to assess the computational performance of our method. The running speed of our method is recorded in Table 9 in Appendix C.4, which provides insights into its efficiency.
> From Table 9, it can be observed that our method, once trained, does not require extensive computation resources for evaluating new systems. The inference process is efficient compared to learning-free methods, which need to optimize from scratch and evaluate the resilience objective function multiple times. For large graphs with 1000 nodes and a maximal number of rewiring being 200, our method takes around 60 seconds, whereas learning-free methods may take 1000-2000 seconds. This demonstrates the computational efficiency of our method during the testing phase.
>
> Regarding the scalability of FireGNN, the time complexity is determined by the number of graph snapshots in the filtration, denoted as K, and the computation required for regular GNNs and aggregation. In Section 5.3, we have added suggestions for selecting the filtration order K in practice. When computational resources are limited, K can be set to 1 or 2 for large graphs, which helps manage the computational complexity while still capturing important temporal information. Therefore, the time complexity of FireGNN is manageable for large-scale networks.
>
> Overall, we believe that our proposed method strikes a good balance between computational efficiency and resilience optimization, as demonstrated by the experimental results provided in the paper.
>
> > Unclear Failure Scenarios: The paper presented the proposed methods in a very positive light, showing they outperform existing methods significantly. However, no method is foolproof, and it is essential to understand the limitations and potential failure points of the proposed approach. The absence of a discussion on potential limitations or failure scenarios might create a biased perspective towards the method's overall robustness. Such a discussion is important because it helps users and researchers understand when it might be unsuitable to employ the proposed methods, what issues they may encounter, and what improvements could be made in the future. For instance, how would the method perform in networks with a specific structure that makes edge rewiring difficult or in networks experiencing dynamic changes? Are there certain types of networks where the performance of the method might decline significantly? Including such discussions would provide a more balanced and comprehensive view of the proposed methods.
>
> In our revised paper, we have included a dedicated section, Section 5.5, that discusses the limitations and future work of our method. This section aims to provide a more balanced and comprehensive view of the proposed methods.

---

### Comment · Reviewer_PxTN · 2023-06-28
**Starting the discussion**

Dear all,

From reading your reviews, I get the impression that the primary criticism centres along the lines of (1) **clarity** and (2) **lack of real-world examples**.  It is my impression that point (1) can be addressed quite easily in a revision (I also raised some issues related to that in my review); for point (2), it would be relevant to understand the strictness of the reviewer's comment here. Personally, I think the paper is well-positioned in terms of experiments—my main issue with the experimental section concerns additional details, which can easily be supplied in my opinion.

---

> ### Author Response · Authors · 2023-07-03
> **Thank you for reviewing our paper. We have uploaded a revised version**
>
> Dear Reviewer,
>
> We express our sincere gratitude for your review of our paper and for providing us with numerous constructive comments. Your insightful suggestions have significantly contributed to the improvement of our work. We have taken each of your comments into careful consideration and have made the necessary revisions, which are clearly marked in the revised version of the paper using blue fonts.
>
> In the following sections below, we provide detailed responses to each of the questions raised by the reviewers. We hope that our explanations adequately address your concerns. If you have any further inquiries or require additional clarification, please do not hesitate to let us know. We will be happy to answer them.
>
> Kind regards,
>
> the Authors

---

> ### Comment · Reviewer_NdKT · 2023-07-22
> **My concerns are resolved**
>
> I have carefully read the detailed responses from the authors, I think my concerns are resolved, and the revision in the paper is appreciated. I'm leaning towards accept.

---

### Decision · Action_Editors · 2023-08-23

**Recommendation:** Accept as is

**Comment:**

Reviewers were generally unanimous in appreciating the importance of the tackled problem as well as the novelty and effectiveness of the proposed approach.  Some concerns were raised, in particular with regards to presentation and clarity, but these have been effectively addressed by the authors during the discussion period.  In light of this, I recommend accepting the manuscript as is.

**Audience:**

The paper is of interest to significant portions of the TMLR community

**Claims And Evidence:**

Claims made in the paper are properly supported by evidence

**Resubmission Of Major Revision:**

The authors may consider submitting a major revision at a later time.